# Development of Advanced Positioning Techniques of UWB/Wi-Fi RTT Ranging for Personal Mobility Applications

**DOI:** 10.3390/s24237520

**Published:** 2024-11-25

**Authors:** Harris Perakis, Vassilis Gikas, Günther Retscher

**Affiliations:** 1School of Rural, Surveying and Geoinformatics Engineering, National Technical University of Athens, 157 80 Zographos, Greece; hperakis@central.ntua.gr (H.P.); vgikas@central.ntua.gr (V.G.); 2Department of Geodesy and Geoinformation, TU Wien—Vienna University of Technology, 1040 Vienna, Austria

**Keywords:** ultra-wide band (UWB), Wi-Fi (wireless fidelity) RTT (round-trip time), integration, performance analysis, fusion, range estimation, localization, pedestrian navigation

## Abstract

“Smart” devices, such as contemporary smartphones and PDAs (Personal Digital Assistance), play a significant role in our daily live, be it for navigation or location-based services (LBSs). In this paper, the use of Ultra-Wide Band (UWB) and Wireless Fidelity (Wi-Fi) based on RTT (Round-Trip Time) measurements is investigated for pedestrian user localization. For this purpose, several scenarios are designed either using real observation or simulated data. In addition, the localization of user groups within a neighborhood based on collaborative navigation (CP) is investigated and analyzed. An analysis of the performance of these techniques for ranging the positioning estimation using different fusion algorithms is assessed. The methodology applied for CP leverages the hybrid nature of the range measurements obtained by UWB and Wi-Fi RTT systems. The proposed approach stands out due to its originality in two main aspects: (1) it focuses on developing and evaluating suitable models for correcting range errors in RF-based TWR (Two-Way Ranging) technologies, and (2) it emphasizes the development of a robust CP engine for groups of pedestrians. The results obtained demonstrate that a performance improvement with respect to position trueness for UWB and Wi-Fi RTT cases of the order of 74% and 54%, respectively, is achieved due to the integration of these techniques. The proposed localization algorithm based on a P2I/P2P (Peer-to-Infrastructure/Peer-to-Peer) configuration provides a potential improvement in position trueness up to 10% for continuous anchor availability, i.e., UWB known nodes or Wi-Fi access points (APs). Its full potential is evident for short-duration events of complete anchor loss (P2P-only), where an improvement of up to 53% in position trueness is achieved. Overall, the performance metrics estimated based on the extensive evaluation campaigns demonstrate the effectiveness of the proposed methodologies.

## 1. Introduction

The tracking of vehicles, pedestrians, and assets in any platform mode is crucial for supporting various societal functions, spanning from personal mobility services to safety and well-being. With the rise in affordable “smart” devices, the provision of high-quality positioning services has become essential for a wide range of wirelessly connected devices. While Global Navigation Satellite System (GNSS) technologies offer a global solution of satisfactory accuracy outdoors, their performance degrades significantly in hybrid environments and becomes impractical indoors [1,2].

This paper serves two primary objectives. Firstly, it focuses on developing a framework for characterizing and modeling RF-based ranging observables derived from disparate radio localization technologies using empirical models. Secondly, it aims to design a methodology for collaboratively localizing groups of autonomously moving nodes (pedestrians) indoors. This is achieved using data derived between rover units and on-site fixed devices (i.e., Wi-Fi access points) as well as among neighboring rovers.

This is accompanied by the development and implementation of associated assessment procedures for performance evaluation. The paper evaluates the performance of radio localization technologies through controlled experimental trials. Statistical analysis enables the characterization of range observable errors generated by highly accurate Ultra-wide Band (UWB) and less accurate Wi-Fi Round-Trip Time (Wi-Fi RTT) technologies. Analysis offers valuable insights into each technology’s capabilities at various operating conditions (i.e., static or dynamic) and in the presence of obstacles commonly found indoors (walls, pedestrian shadowing, etc.). Furthermore, the paper proposes alternative techniques for range error mitigation, including the development and evaluation of suitable empirical correction (linear and spatial) models.

The methodology applied for the collaborative localization of a group of users (e.g., pedestrians) leverages the hybrid nature of the range measurements obtained by UWB and Wi-Fi RTT (Round-Trip Time) systems. Firstly, the proposed algorithm calculates the standalone position of the moving nodes using the existing communication infrastructure (Wi-Fi RTT) for Pedestrian-to-Infrastructure (P2I) ranges. Subsequently, the localization solution obtained in the P2I step is combined with ad hoc UWB Pedestrian-to-Pedestrian (P2P) ranges and orientation observables (i.e., loosely coupled filtering scheme) to serve as a baseline for implementing a distributed collaborative position solution. The collaborative positioning system yields an improvement in accuracy and availability compared to the absolute P2I localization solution, while reducing the need for extensive infrastructure equipment. Position availability refers to the percentage of time during for which the positioning terminal delivers a position solution. The quality metrics of the localization algorithms are assessed with field as well as simulated datasets generated using in-house developed software.

The paper is structured as follows: After the introduction in Section 1, Section 2 presents, after the general aspects in Section 2.1, the research questions and objectives and the methodology in Section 2.2 and Section 2.3, respectively. Section 3 then introduces the range-based CP system, where firstly in Section 3.1, the range errors and their mitigation are identified followed by a description of the architectures and algorithms for CP in Section 3.2. In the following, Section 4 deals with the introduction of the different employed range correction models, including the correction model’s development steps in Section 4.1, the detailed discussion of these models in Section 4.2, the orientation-assisted range correction models in Section 4.3, and then their validation in Section 4.4. Section 5 then analyzes the position computation algorithms with the correction models adopted for the UWB and Wi-Fi RTT range observable internal accuracy in Section 5.1 and the Kalman Filter formulation for distributed CP in Section 5.2. Section 6, on data collection and error mitigation, includes Section 6.1, Section 6.2 and Section 6.3 on the description of the test data and equipment employed as well as the properties of the indoor field test campaigns and the range error mitigation for both UWB in and Wi-Fi RTT measurements. Although measurements in outdoor environments have been carried out in this study, the paper focuses more on the results obtained indoors in Section 7, where the position solution estimation is thoroughly discussed in Section 7.1 and Section 7.2 with the localization solutions obtained using field and simulated data, respectively. Finally, Section 8 concludes the paper and provides a detailed outlook on future work which will be carried out by the authors of this contribution.

## 2. Motivation of This Research

### 2.1. General Aspects

The motivation for this research work stems directly from the recent advances in IT (Information Technology) and MEMS (Micro-Electromechanical Systems) technologies and their global adoption in contemporary smartphone and PDA (Personal Digital Assistant) devices. In this era, the potential of recently introduced RF ranging technologies is significant, especially when integrated with inertial sensor data. The driver for undertaking this research study is the recent advances and potential in the following open research areas:RF (Radio Frequency) ranging technologies of high accuracy in the UWB spectrum have recently been introduced to commercial smartphone devices. The continuously decreasing size and cost of UWB augurs their wide adoption by the industry in the near future; it is forecasted to grow from USD 1.82 billion in 2024 to USD 4.08 billion by 2029 [3].The recently introduced Wi-Fi RTT technology can easily offer P2I ranging services of medium nominal accuracy (0.5 m) as part of the default web access functionality in a widespread and seamless manner [4]. Its fast adoption by the smartphone industry [5] dictates the pressing need for in-depth studying of the ranging capabilities and limitations, as well as developing methods towards improving the derived localization solution.The RF-based TW-ToF (Two-Way Time of Flight) ranging principle introduces inherent and condition-specific (bound to varying devices and environment conditions) inaccuracies of a variable nature [6,7,8,9]. As a result, the need for thoroughly studying its error budget, and the development of alternative error mitigation techniques capable of adapting in different environments, is clearly evident.The combination of UWB and Wi-Fi RTT technologies may offer increased coverage and flexibility of indoor positioning solutions by combining pre-existing P2I ranging infrastructure with ad hoc P2P ranging [10]. This ubiquitous ranging setup utilizing these complementary technologies has still not been extensively studied [11].The goal in real-life applications leans towards reducing the number of infrastructure (anchor) nodes in order to lower procurement and maintenance costs [12,13,14] further enhanced by approaches utilizing Reconfigurable Intelligent Surfaces (RISs) or metasurfaces [15,16]. Therefore, the proposed positioning solutions should incorporate flexible architectures utilizing optimally both the available anchor points and moving nodes. Therefore, a study addressing the alternative setups of available anchors, moving nodes, and their geometry distribution in selected operational scenarios is expected to provide useful insight for designing indoor positioning system (IPS) installations.A crucial problem to be addressed in P2P range-based collaborative decentralized positioning architectures is the mitigation of errors induced due to unknown correlations among the communicating nodes. Relevant studies that have concluded on stable Position Velocity and Timing (PVT) solutions are employing node classification concepts—for instance, using primary nodes equipped with multiple sensors and absolute position knowledge in order to provide internodal positioning input to secondary P2P-only rovers [17,18,19]. This approach requires a multilevel rovers design, whereas the overall system operates with reduced flexibility due to its dependence on the continuous operation of high-cost rovers. Therefore, as long as the mitigation of the propagated position errors for collaborative P2P range-only positioning algorithms is not addressed, the provision of a robust and scalable real-time solution still remains an open issue.

### 2.2. Research Questions and Objectives

The main research questions to be answered, and the primary and secondary objectives of this research can be summarized as follows:Primary Objective 1: To develop and test a methodology for identifying and mitigating errors in TWR (Two-Way Ranging) RF range observations;Primary Objective 2: To develop and test a robust, RF range-based positioning approach for groups of pedestrians walking in dynamic environments, considering the hybrid nature of TWR measurements;Secondary Objective 1: To establish and implement a unified Quality Control (QC) framework for the assessment of the correctness and efficiency of the proposed solutions.

For the first research objective, the initial sub-objective involves conducting methodical field tests in controlled environments to examine TWR RF range errors. These tests aim to retrieve valuable feedback using actual data. This part of the investigation aims at a thorough statistical analysis and characterization of the nature of errors in range observations produced using the UWB and Wi-Fi RTT technologies. This investigation is expected to provide useful insight concerning the technologies’ capabilities in varying operational conditions (e.g., static or kinematic operation) and against obstacles interaction (e.g., through-wall operation and pedestrian shadowing) encountered commonly indoors. Based on the findings of range error characterization, alternative range error mitigation techniques are suggested. This sub-objective aims at developing, implementing, and evaluating suitable range error correction procedures based on specific empirical range error correction models. These models are built to be valid in environments with common characteristics (i.e., different building layouts and/or materials), data availability setup (i.e., transceivers number and installation geometry), obstacles (i.e., pedestrians and walls), as well as different RF technologies (i.e., UWB and/or Wi-Fi RTT) in order to propose a solution providing robust range error correction.

The second topic is addressed: Firstly, the proposed algorithm computes the standalone position of the moving nodes in question aided by the existing communication infrastructure. For this purpose, appropriate localization algorithms have been studied for handling the Pedestrian-to-Infrastructure (P2I) ranges obtained by Wi-Fi RTT. The proposed localization engine implements the range-error correction techniques for mitigating inherent Wi-Fi RTT inaccuracies while providing real-time functionality. The second part of the localization approach integrates the range measurements along with the location of the neighbor moving nodes obtained in the previous step as a basis for the development of the distributed collaborative positioning (CP) solution. This goal resides in ad hoc Pedestrian-to-Pedestrian (P2P) UWB ranging technologies in order to support and extend the positioning availability provided by the P2I absolute pedestrian positioning. Overly, the CP engine should be also capable of integrating range error mitigation models whilst ensuring its ability to operate in real time. Finally, in order for the proposed CP engine to be adaptive enough to compensate for dynamic conditions, it is designed to optimally use the P2I and P2P range information. The main goal is to develop a robust algorithm able to accept a different number of anchors (P2I) while minimizing the effect of propagated P2P correlation-induced positioning errors. Appropriate algorithms are proposed for handling the correlated and uncorrelated errors among communicating inter-ranging moving nodes. It is noted that the proposed CP algorithm adheres to inertial measurements via low-complexity range/IMU fusion in order to compensate for the short-term P2I range unavailability.

The first sub-objective refers to the evaluation of the proposed range error correction models under varying conditions, for different datasets collected using both TWR technologies under consideration. The field-testing procedures are repeatable and are able to be performed in varying locations. Through the implementation of extended, dedicated field tests, a detailed analysis and evaluation of the different correction models should lead to concrete proposals suitable for each RF ranging technology.

The second sub-objective refers to the detailed and extended testing and assessment of the proposed suite of positioning algorithms using real and simulated data. The design and implementation of dedicated field experiments enable the acquisition of complete datasets for testing the proposed algorithm and for fine-tuning the subsequent, extensive simulations tests. Regarding simulation testing, the development of a ranging and orientation data generator based on simulated or real trajectories enables the evaluation in a controlled and repeatable manner. The generated data simulate the quality characteristics of the TWR technologies in order to provide a straightforward assessment based on controlled reference data.

### 2.3. Research Methodology

The research methodology followed in this work consists of three distinct but inter-related implementation steps:Range measurements calibration/correction phase;Positioning algorithms development;Quality Control of the positioning engine utilizing real and simulated datasets.

The first key item includes (1) the pre-analysis stage, (2) the correction models development, (3) the error mitigation and models validation, and the (4) kinematic range error correction.

Initially in step (1), the raw RF range observables need to undergo a preliminary statistical analysis in order to characterize the TWR technologies’ behavior and pave the road for the following analysis steps.

In the correction models’ development, the raw data collected in the previous stage undergo a refinement process. The statistical metrics obtained are used in order to guide data grouping, outlier identification, data exclusion, or even the repetition of data collection campaigns. Specifically, the empirical range error models produce both radial (1D) and spatial (2D) ranges before the implementation on real data. Additionally, the models are combined with available RSS (Received Signal Strength) indicators as well as user orientation information enabling investigation of the environmental effects.

For the error mitigation and models validation, the developed ranging errors models are applied on real datasets for static conditions, providing initial feedback regarding model performance. In order to ensure unbiased estimation, the calibration evaluation is performed on data collected for validation purposes and not on the initial data collected for the generation of error models. Moreover, the evaluated range error models are implemented on data referring to well-defined operational conditions (e.g., room type and geometry), providing the evaluation of the error models based on real reference data.

And finally for step 1, the kinematic range error correction is applied. Ultimately, the range errors mitigation process is of importance for kinematic positioning sessions. The dynamic characteristics of kinematic data provide the most demanding conditions for TWR range error model validation due to varying environmental effects. For validating the reliability and robustness of the models, the performance evaluation is performed utilizing sets of test trajectories that cover the majority of the available testbed areas.

The second key item of the positioning algorithms development includes (1) the tuning of the positioning filter, (2) the collaborative positioning algorithms, and (3) the cross-correlation effect mitigation.

For the tuning of the positioning filter, the positioning engine’s core is the Kalman Filter (KF), which is developed and optimized to handle UWB and Wi-Fi RTT raw observables. During this development process, both process and measurement noise is fine-tuned in an optimal manner. The tuning is based on the statistical characterization performed for both TWR technologies in the previous development step. Azimuth information obtained by onboard IMU is included within the KF in order to compensate for epochs of gross ranging errors or ranging measurables unavailability.

Then, the collaborative positioning algorithm is developed for providing localization in a robust, scalable, and self-contained manner. For that purpose, a distributed architecture is selected. Considering that the foundation of the system relies on the utilization of Wi-Fi RTT for P2I ranging and UWB for P2P ranging, the mathematical models and algorithms should be designed and developed accordingly. Operational elements for each technology, such as sampling rate, data formatting, and communications scheduling are taken into account. At this stage, the collaborative positioning operation is limited to one rover utilizing multiple ranges from neighbor anchors and rovers, omitting the unknown correlation effects on positioning errors during multiple rovers’ localization.

For the cross-correlation effect mitigation, an extension of the developed CP algorithms is proposed based on the approach of the Split Covariance Intersection Filter (SCIF), which can be implemented as a variation of KF. The SCIF approach incorporates the cross-correlation in errors occurring due to the relative measurements among collaborating nodes. The proposed approach utilizes range-only relative measurements and the communicated position state of the rover. As this is a range-based approach, the developed filter also incorporates an adaptive KF feature for compensating for an abrupt orientation change. Since absolute positioning is provided by the P2I ranging technology, the maximum expected positioning performance is bound by the ranging quality of Wi-Fi RTT observables.

The third key item of the Quality Control (QC) of the positioning engine utilizing real and simulated datasets includes (1) the field testing campaigns, (2) the raw observables simulator, and (3) the performance evaluation.

The selected testbed areas in the field testing campaigns need careful selection in order to ensure typical environmental conditions that are required at the evaluation stage of the proposed approach.

For the evaluation, a raw software simulator for ranges and heading is designed and developed to aid in the development, testing, and field-testing design of the CP algorithms. The software is based on the configurable simulated trajectory data of multiple simultaneously roving nodes. Modular errors can be introduced for the range measurements in order to simulate the different quality of the TWR technologies at hand. Moreover, the feature of dynamic anchor availability enables the study of simulated obstruction effects commonly present in indoor environments. The performance evaluation statistics are estimated utilizing the preconfigured reference trajectories.

In the performance evaluation, the overall assessment of the proposed approach is performed assuming standard quality metrics for relevant PVT-reliant applications (e.g., trueness, accuracy, availability, and Dilution Of Precision) in varying operational conditions. The estimation of the statistical measures takes place for varying range error correction models, leading to an overall evaluation of the proposed range error mitigation approaches. Finally, the effects of varying motion characteristics of the rovers are examined, as different dynamics directly affect the localization engine performance.

## 3. Range-Based Collaborative Positioning (CP)

This section provides a brief background summary on the basic techniques, the measuring principles, and mathematical fundamentals for indoor CP determination using range observables. Inter-nodal ranging may refer both to range measurements originating from roving nodes to static anchors as well as between roving nodes. A description of the usable positioning techniques and methods as well as optimization algorithms in positioning and sensor fusion is omitted, as it can easily be found in the literature, e.g., in the review paper of one of the authors of this contribution [20].

### 3.1. Range Error Identification and Mitigation

In order for a positioning technique to produce an optimal solution, it is important that the raw observables (ranges, directions, etc.) have undergone exhaustive preprocessing to mitigate gross and systematic errors; see [21]. In particular, in the indoor environment, which is characterized by NLOS (Non-Line of Sight) conditions and severe signal multipath, the raw range observables can be of low quality. Extensive research is currently being undertaken by many research groups worldwide studying the nature of RF-based range errors and modeling their behavior, aiming at minimizing their effect on the final position solution [22,23]. Moreover, the combined effects of NLOS conditions, multipath, signal attenuation and scattering in wireless positioning systems further deteriorate the position quality, as it is subjected to travel from the transmitter to the receiver through multiple paths.

Considering that NLOS conditions represent a major challenge for indoor RF-based positioning applications, various research efforts have focused on methodologies aiming at mitigating the NLOS effects. As in real-life applications, the existence and severity of NLOS conditions is a priori unknown; a research approach should aim at characterizing signals as LOS or NLOS. Thereby, if a signal is identified as a LOS one, then no prior action is required, contrarily to signals detected as NLOS ones. The latter undergo dedicated preprocessing techniques for mitigating the respective errors [24,25]. The distinction between LOS and NLOS observables can rely either on sequential range estimation for outliers’ thresholding or on channel statistics [26]. Relevant studies suggest that the non-Gaussian distribution nature indicates an obstacle when working with KF algorithms since they assume that the measurement errors follow a Gaussian distribution [27,28]. Subsequently, for the indoor cases of a mainly non-Gaussian TWR observation nature, it is expected that they are prone to position quality instability due to model assumptions. Attempts to overcome this limitation usually rely on the adoption of non-linear measurement error models, leading usually to particle filters (PFs) [29,30]. However, a PF solution asks for increased computational complexity, which is not easy to support by handheld, low-cost indoor positioning systems. Alternative approaches include realizations of hybrid KF implementations based on pseudo-position measurements that can handle non-Gaussian error models [31]. While they offer reduced computational complexity compared to PF, they still require increased processing power compared to traditional KFs. An alternative approach for handling the non-linear nature of the range error observables indoors is via a Gaussian Mixture (GM) filter type. Such filters can handle error distributions with multipeaks [32,33]. In effect, they apply multiple Gaussian models to approximate the complex nature of the transmitted signals; however, it is crucial to identify and use the optimal number of Gaussian components to avoid unnecessary computational complexity. While this approach offers increased positioning accuracy for highly noisy measurements, its computational complexity increases dramatically for multinode, range-based positioning. It is noted that while the KF approaches reach their limit in highly non-linear cases, still the EKF offers a viable alternative when handling moderately non-linear error models due to their computationally efficient architecture [34,35].

Empirical RF range error models, on the other hand, rely on the systematic collection of real range observables to extract meaningful statistics that adequately describe their nature and extract range variation behavior that might be encountered during real-life localization applications. Examples of empirical modeling of RF-signal for localization include the approach introduced by Li et al. [6] that relies on an asymmetric, double exponential ranging error distribution model. The error model is formulated through fitting real data, whereas an extension of further tuning the suggested model using range-based parameters is proposed. In the work of Jing et al. [36], a Ranging Quality Indicator (RQI) is established based on UWB signal characteristics paired with the corresponding ranging error used to train a machine learning (ML) algorithm. In this approach, the algorithm produces a set of RQI values in real time and dynamically assigns weights to the range measurements in a UWB/IMU particle filter. In a study by Koppanyi and Toth [37], the original UWB range histograms are found to present multiple peaks attributed to multipath effects. To this effect, a Maximum Likelihood Estimator (MLE) is used for selecting the ranges with the highest probability of true values based on a comparison against the lateration-derived coordinates. Moreover, other empirical error models use range- and position-dependent corrections produced using curve-fitting approaches on real data as illustrated in Figure 1.

In a research study by Toth et al. [38], range error calibration is implemented based on a grid of calibration points used for the generation of an ad hoc model. In this approach, the calibration values are used for the 2D linear interpolation forming the calibration function. In the work of Ledergerber and D’Andrea [7], a Sparse Pseudo-input Gaussian Process is trained using the known relative antenna pose (angle) and the error computed using the fixed distances between UWB nodes. The objective is to build an error prediction model that will be utilized in Kalman Filter-based UWB positioning. Regarding the field-testing setups followed for UWB range error analysis and identification, different approaches exist depending on the testing scope. On one hand, when extensive characterization of the experimental area needs to be conducted, the tests are focused on the collection of extensive datasets for the dedicated site. For instance, Li et al. [6] perform a series of static indoor field tests for estimating the range error values using multiple anchor nodes and mobiles. The area’s concrete and steel walls result in predominantly NLoS conditions, while the entire sets of data (LOS, multipath and NLOS) are analyzed together simultaneously with the error models generation for improving positioning performance. On the other hand, for generalization purposes, it is common practice for experimental implementation to take place on test sites featuring different characteristics. In the work of Toth et al. [38], tests are conducted under various observation conditions—i.e., a combined outdoor open area, a forest environment, and indoors. The different environment conditions indicate the varying effects on UWB positioning. Subsequently, the error calibration process is based on known calibration points forming a grid.

### 3.2. Collaborative Positioning

An increased interest towards the development of CP approaches is apparent in the recent literature; nevertheless, the concept is not a new one [39,40]. The increased motivation for CP stems both from the technological developments for utilizing optimally Peer-to-Peer (P2P) communication as well as from the need for minimizing the costs of the permanently installed infrastructure (i.e., anchor RF transceivers) used by traditional RF-based positioning systems. In many cases, P2P communication between nodes is based on technologies that can also offer relative ranging such as Wi-Fi, UWB, and Bluetooth [41,42]. In this regard, CP implementations make use of these technologies both for application-specific data transmission as well as for supporting localization needs. This section presents a short description of CP approaches, their architecture, and the most prevailing CP algorithms, as well as an overview of the implemented CP approaches with varying operational conditions.

The network architecture of a CP system can either be a centralized or distributed one [43]. These two different architectures are illustrated in Figure 2. In a centralized architecture [6,17,44,45], as the name suggests, the positions estimation is performed centrally by a localization engine typically located at a control center that collects data from all the remote nodes. Central processing translates at increased processing power considering that state (position, orientation, and velocity) computation of all nodes in the network is undertaken by a single processing engine. Naturally, as the information from all nodes in the network needs to be transmitted to the central unit for the estimation to be complete, this approach also leads to increased communication requirements. In addition, as CP systems rely on a single, central engine processing unit with finite processing and communication capabilities, the expansion for increasing (scalability) the supported number of nodes faces crucial limitations. Notwithstanding an appropriately designed and implemented centralized CP engine offering high-accuracy pose estimation for all nodes and inter-nodal state correlations, it suffers decreased robustness. The dependence on a single, central processing engine for continuous operation results in high-probability operational malfunctions. On the other hand, distributed CP architectures depend on their ability to self-estimate nodal positions based on the measurements and information collected within the CP network [19,36,46]. Practically, in order to achieve this goal, each node in the network needs to be equipped with a portable processing unit and a certain communications infrastructure. This translates to decreased processing capabilities and therefore more stringent limitations on the amount of received data that can be supported, and by extension, the accuracy capabilities of the overall system. A useful trade-off is the ability to operate with limited communication among the collaborating nodes as well as to easily integrate additional collaborating nodes, resulting in a highly scalable system. Perhaps the most crucial weaknesses of the distributed CP approach are their inability to maintain inter-nodal correlation at the network level leading to decrease the mitigation of inter-dependent errors. Table 1 summarizes the strengths and weaknesses of the centralized and distributed CP architectures.

As the overall motive of this study on utilizing collaborative localization relies on the ability of independent mobile nodes to handle independent positioning information and relative measurements from neighboring nodes, the interest of the current research is focused on the application of distributed localization. The distributed collaborative localization problem for nodes performing relative range measurements can be addressed using four main positioning algorithms: (1) the non-linear Kalman Filter; (2) the particle filter; (3) the belief propagation; and (4) the covariance intersection. As the first two are well known, in the following, only the third and fourth algorithms are briefly elaborated on.

Belief propagation algorithms rely on factor graphs, and particularly on the well-known Sum Product Algorithm over Wireless Network (SPAWN), which is an inherently cooperative localization approach. It relies on the exchange of messages for each node in the network to determine its a posteriori distribution given all the available measurements; see [47]. Despite being able to provide highly accurate results, the SPAWN algorithm also suffers high computational complexity as well as requiring a specialized configuration for handling loopy networks (i.e., the “outbound” ranging observable affects the consequent “inbound” observables) [48,49].

When fusing information among neighboring nodes within a CP network, a highly challenging task is the mitigation of accumulated inter-dependent errors, that is, the computation of state correlations among cooperating nodes that utilize shared positions and relative range information. Clearly, inter-nodal correlation may lead to non-converging positioning solutions if not accounted for. A Covariance Intersection Filter (CIF) approach attempts to mitigate the effect of unknown correlations by combining multiple estimates of state variables in the form of means and covariances, assuming that regardless of whether their correlation is unknown, the variables are always correlated [50,51]. The extension of the CIF concept to the Split-Covariance Intersection Filter (SCIF) aims to support this generalization of the correlation by splitting dependent (i.e., position and variance) and independent information (i.e., ranges and error) before the covariance intersection estimation [52]. As the SCIF may be implemented in the form of a modified KF, low complexity is ensured for multiple node position estimation, offering an attractive alternative for CP networks. A limitation of SCIF is identified in the requirement for the relative position to be known among cooperating nodes for successful solution convergence.

Table 2 summarizes the main strengths and weaknesses for the EKF, PF, SPAWN, and CIS/SCIF algorithms.

## 4. Range Correction Models

Section 4 presents the methodological framework for the development of range correction models. Based on the statistical measures obtained using UWB and Wi-Fi RTT observables, we propose distinct correction models and describe their respective implementation steps. Model validation procedures are established, and the associated developed software is presented.

### 4.1. Correction Models Development Steps

The methodology followed for the design, development, implementation, and evaluation of the TWR range correction models relies on distinct steps as described in the following. These are developed as follows:Statistical characterization of range errors: At a first stage and before any range error modeling is applied, the raw TWR measurements undergo preliminary statistical analysis. For this purpose, a number of specifically designed experiments take place using an accurately surveyed testbed. In this regard, the statistics of the raw ranges carry useful information supporting the follow-up step of developing the data-driven range error models.Empirical range error models development: Prior to defining a range error model, the statistical metrics of the raw data obtained in the previous step are evaluated to assist in data grouping and data exclusion, or even dictate further data collection. The generation of error models resides on data-driven optimization techniques using regression analysis tools (best-fitting curves, interpolation, etc.). Statistical evaluation of the models before the implementation on real data enables the identification of potential gross deviations and data over-fitting.Error mitigation: This step includes implementation of the error models on real range data. Obviously, in order to obtain an unbiased evaluation, data correction refers to data collected only for validation purposes, excluding all data used for building the error model. By design, and in order for the error models to be efficient, they are classified to suit different operational conditions, usually by room type and geometry. Validation of the efficiency of error modeling is undertaken using a suitable subset of static reference data.Kinematic positioning: The final step of the range correction methodology concerns the error model performance assessment at the operational level. The dynamic character of the kinematic data provides the most demanding conditions for the TWR range error model validation due to the varying environment and user kinematics. For validating their reliability and robustness, a set of test trajectories is built that covers the entire testbed area.

Figure 3 depicts the overall range correction procedure adopted in this study.

Preliminary analyses suggest that TWR measurements do not necessarily follow a normal distribution indoors for reasons relating to multipath and through-material propagation effects. Therefore, the selection of a suitable statistical value is suggested. Figure 4 and Figure 5 show typical histograms of range datasets collected for the case of UWB and Wi-Fi RTT devices, respectively, for indoor environment conditions. Clearly, the histograms in the two figures indicate that the mean value cannot adequately represent the range sample. Moreover, while the median value provides a somehow improved index using the Empirical Probability Density Function (EPDF) for defining, the maximum likelihood value (EPDFmax) provides an optimal fit. The EPDFmax values need to be estimated, given that the respective histograms may not be utilized as a probability measure since they consist of discrete values (bins) that result in varying shapes based on the different bin sizes. The EPDF is estimated using kernel density estimation. It is crucial to select appropriate kernel bandwidth values, as larger bandwidth values smooth out the relevant peaks of EPDF, whereas for very small values, the remaining overall fluctuation hinders correct EPDFmax value estimation [23]. The empirically estimated kernel bandwidth value of 0.005 results in a good fit for the UWB data using the P410 module (Time Domain^©^, Huntsville, AL, USA), whilst the selection of a kernel bandwidth value of 0.02 results in a good fit for the WILD module (Compulab^©^, Yokne’am Illit, Israel) Wi-Fi RTT ranges.

### 4.2. Range Correction Models

Following previous studies, the correction process for TWR data could be based either on empirical radial corrections, applying a least squares line fit to the range deviations as a function of the distance [23], or using a 2D range deviations plane fit [38]. In this study, we examine both approaches and extend the examination to WILD Wi-Fi RTT data in order to select the appropriate correction technique that suits the corresponding dataset.

#### 4.2.1. Radial (1D) Fitting Model

The development of a radial (1D) range correction model assumes the collection of TWR data at known (reference) distances using the RF devices of interest. For each pair of RF-ranging devices, a set of range measurements is collected to estimate their statistics and their deviation from the reference value. Hence, the correction value computes the difference between the one-way uncorrected measurement from the true (reference) distance as follows:(1)$$range_{correction}=range_{true}-range_{measured}$$

Obviously, the ranges correction reflects the operational characteristics of the RF devices and the observation conditions applied in the area zone between the RF devices in use. The correction values may be estimated for various inter-device conditions in order to examine different environmental effects (i.e., NLOS conditions). The range correction models are realized through curve fitting on field data. Depending on the individual characteristics of the specific TWR technology and environmental conditions, different fit models may apply for each approach. The type correction models usually adopted are the “mean”, the “linear”, and the “polynomial” (second-order polynomial) fit. Figure 6 illustrates examples of various empirical correction models for UWB measurements. Notwithstanding the “polynomial” model appearing to more closely describe the nature of the range correction, a thorough examination is required in order to avoid over-fitting effects. Within this work, the models adopted refer to a linear fitting approach, as it has proved to better describe the collected TWR data, avoiding over-fitting effects. Figure 7 provides a schematic view of the procedure for empirical range correction models generation, outlining the distinct steps.

For the case of a radial (1D) correction model, two variations are considered in this work, a generic linear correction model that covers all examined area, and a segmentation-based linear correction model to improve spatial resolution at a room level. The structure of the segmentation-based approach relies on the distribution of correction points in corresponding rooms.

The first range correction approach (All Rooms Linear Correction, “*arlc*”) produces radial corrections for the complete test area irrespective of the room characteristics, and therefore, no distinction is made between LOS and NLOS conditions. The corresponding range correction equation reads
(2)$$arlc_{i}^{n}=d_{i}^{n}+f_{l}\left(d_{i}^{n}\right)$$
where $f_{l}$ is the linear range correction equation for all rooms for anchor node *n*.

The second range correction variation (Room Linear Correction, “*rlc*”) produces a linear approximation of the correction values individually for each room depending on the continuously LoS or NLoS ranging conditions to specific anchor nodes each time. For instance, considering the case of Figure 7, the correction model for the left room corresponds solely to LOS ranging for anchors A1 and A2, and to NLOS ranging for anchors A3 and A4. The equation describing *rlc* reads
(3)$$rlc_{i}^{n}=d_{i}^{n}+f_{L}\left(d_{i}^{n},j\right)$$
where $d_{i}^{n}$ is the current (*i*) measured range between the roving node and anchor node n, and $f_{L}$ is the linear range correction equation for room *j*.

#### 4.2.2. Spatial (2D) Fitting Model

The generation of the two-dimensional range correction approach is based on the same underlying principle as the 1D approach (see Figure 8). In essence, the differences between the measured and true (reference) distances are used for the generation of a correction database connecting the correction points. In comparison to the linear fitting model, this approach takes into account the spatial distribution of the test ranges in the area of interest. Therefore, this method provides a bi-dimensional correction fit, which accounts for the location of each correction point. In order to cover the entire area of a test site, the correction values are interpolated using natural neighbor interpolation [53], which is based on the Voronoi tessellation method; hence, this Voronoi correction approach is denoted as “*vc*”. For the area found outside the polygons defined by the correction points, linear extrapolation is performed in order to extend the Voronoi correction values. The equation describing vc reads
(4)$$vc_{i}^{n}=d_{i}^{n}+f_{v}\left(x_{i}^{n},y_{i}^{n}\right)$$
where $f_{v}$ is the bi-dimensional range corrections equation for the moving node’s position ($x_{i}$,$y_{i}$) for anchor node n.

### 4.3. Orientation-Assisted Range Correction Models

#### 4.3.1. Orientation Assist

Among the most influencing drawbacks concerned with TWR observables indoors are NLOS effects generated by physical obstacles or the multipath. In an attempt to initially model and consequently mitigate the effect of NLOS conditions in TWR ranges, orientation-assisted range error modeling is conceptualized and evaluated. In this regard, the most influencing factor associated with NLOS conditions for pedestrian indoor positioning is the same pedestrian’s body acting as a live obstacle. In order to examine and evaluate in a systematic manner the user orientation effect in relation to the anchor point, the data collection campaigns’ resolution described before in Section 4.2 is further increased by introducing the collection of discrete ranging datasets for all four cardinal orientations (north, east, south, and west).

According to the *rlc* correction model, this approach generates a linear approximation of the correction values for each orientation. The orientation–linear correction model ("*olc*") is described by the equation
(5)$$olc_{i}^{n}=d_{i}^{n}+f_{or}\left(d_{i}^{n},or\right)$$
where $d_{i}^{n}$ is the current (*i*) measured range between the roving node and anchor node *n*.

Moreover, the expansion of the spatial (2D) correction model in order to include an additional level of detail based on the orientation assist is proposed and can be formulated as the orientation–Voronoi correction model (“*ovc*”) and is defined as follows:(6)$$ovc_{i}^{n}=d_{i}^{n}+f_{ov}\left(x_{i}^{n},y_{i}^{n},or\right)$$
where $f_{ov}$ is the bi-dimensional range correction equation for the moving node’s *n* position ($x_{i}$,$y_{i}$) and for each orientation.

#### 4.3.2. RSS-Based Orientation Selection

In order to apply the correction models discussed in Section 4.3.1 in real case scenarios, the user orientation should be known. Notwithstanding today’s technology (e.g., MEMS IMU) being able to compute user orientation, at this stage, we exercise an autonomous RF-based approach. The proposed approach relies (1) on the provided data of each RF-based conversation, including both TWR observables along with signal quality information (RSS), and (2) on the hypothesis that the main source of RSS fluctuation for an otherwise static rover is the change in orientation due to the imposed NLOS conditions. Therefore, user orientation estimation relies on the comparison of the collected real-time RSS values against those obtained from previously collected RSS values for consequently selecting the appropriate orientation-based correction model. For this purpose, in addition to the linear and bi-dimensional models generated for the TWR measurables, the database is also populated with RSS-based linear and bi-dimensional models that are generated in a similar manner. For the case of the “*olc*” model, the RSS values are employed for generating a corresponding linear model for all anchor–rover pairs with respect to the reported range. These RSS models are then used during the online phase of the range correction algorithm by comparing the reported RSS value with respect to the reported uncorrected range and consequently selecting the closer RSS model. Similarly, for the case of the “*ovc*” model, the corresponding spatial RSS models are generated, and the real RSS values are compared against them in order to select the closer RSSI model and consequently the most, respectively, “*ovc*”-type model. Figure 9 illustrates the outline of the described RSS-based orientation selection approach.

### 4.4. Range Correction Models Validation

In order to evaluate the appropriateness and operational efficiency of the range correction models, certain validation approaches are implemented. At the first stage, correction model validation refers to static ranges, aiming at computing detailed statistical measures whilst at the same time providing initial feedback for adopting a suitable correction model for the kinematic case. The second stage deals with the model validation process intended for kinematic positioning, specifically for evaluating range error mitigation effects under realistic positioning scenarios.

#### 4.4.1. Internal and External Parameters Affecting TWR Quality

Due to the inherent characteristics of the TWR observables and indoor environment conditions, which are of prime interest in this work, several factors need to be accounted for at the model validation stage. The internal factors effect refers to the varying setups that the TWR sensors may provide to the user, such as different signal transmission configuration values and sampling rate. The choice of the signal transmission configuration parameters such as the signal bandwidth or Pulse Integration Index (PII) affects ranging performance. Specifically, variations in signal configuration might provide the ability to acquire effectively range measurements over long distances and in return operate in lower sampling frequencies. Moreover, different recording bandwidths may provide variable ranging repeatability (i.e., precision) and multipath effects or NLOS resilience. Additionally, the choice of sampling rate values directly affects the positioning solution performance since, for example, a low sampling rate may hinder the ability to track motions of higher dynamics. On the other hand, a very high sampling rate might impede the localization engines of the roving nodes network, as it requires higher processing power in order to manage the increased data throughput. External effects refer to variations in the environmental conditions when performing TWR positioning. The indoor environment complex geometry, the presence of surrounding obstacles (static or mobile), and a user body acting as the main source of NLOS, are some of the determinant external factors. In addition, RF signal attenuation, scattering, and fading need to be accounted for and evaluated within a validation procedure. The different TWR technologies adopted in this research are expected to provide somewhat varying performance in varying environmental setups. Therefore, a detailed analysis takes place in order to gain insight that will facilitate subsequent experimental evaluation of positioning using a combination of the technologies. The NLOS being the main ranging quality degradation effect is examined using both through-the-wall TWR observables as well as the user’s body in a controlled and repeatable manner.

#### 4.4.2. Validation Procedure of the Static Range Correction Model

The validation of the static range correction model presupposes a series of suitable range datasets collected at different observation distances. This enables the statistical characterization of the raw observables, leading to conclusions about the performance of the correction models. The static validation datasets are collected in the same environment as the correction datasets since the ad hoc error correction models are suited for similar environmental conditions. Notably, performance assessment of the range correction models in variable environments exceeds the scope of this research. Naturally, in order to reach unbiased model assessment, the evaluation of the validation datasets is performed on data collected specifically for validation purposes and not on those collected for error modeling. The number of validation points selected ranges between 30 and 40% of the total datasets points, which is adequate for providing reliable evaluation results. The field procedure includes range observation in a 1D setup from an anchor point to a rover placed sequentially at increasing distances along a corridor-like geometry. Also, the procedure may be performed in a 2D setup. In this scenario, the observation points are spatially distributed throughout the area of interest, and the corresponding Euclidean distances are computed based on the known anchor coordinates.

At the implementation stage, the radial and spatial correction models and associated software are implemented. Subsequently, the corrected ranges are cross-compared against the nominal distances resulting in a statistical evaluation (i.e., trueness, mean, and standard deviation) for gaining insight regarding the parameters analyzed in Section 4.4.1. Corresponding trueness histograms facilitate the quantitative performance evaluation for each pair. Based on the corrected ranges, the remaining error EPDFmax value may be used for all validation points in order to generate remaining error diagrams in contour or heatmap form.

#### 4.4.3. Validation Procedure of the Kinematic Range Correction Model

Since the aim is to enable a correction model for kinematic (dynamic) range evaluation for real-time applications, the validation procedure needs to be expanded for the kinematic case. This validation mode intends to evaluate the performance of the developed models in a realistic manner, whilst at the same time coping with fewer observables than the static one per rover position, and therefore no detailed statistical measures can be estimated. Usually, the estimation of a reference trajectory indoors relies on the realization of a predefined path along previously established and accurately surveyed points. Positioning performance evaluation relies on the comparison of the estimated trajectories performance using the different correction models. Moreover, the assumptions underlying each model implementation are different, considering that the radial (1D) models rely only on the measured range, while the spatial (2D) models rely on the previously estimated position. This validation step allows for the evaluation of the model implementation in real TWR datasets intended for trajectory estimation. The trajectory quality metrics estimated against the reference trajectory enable the quantitative comparison among varying models.

### 4.5. Developed TWR Correction and Validation Software

Within the scope of designing and implementing a generalized approach for building TWR range correction models, a dedicated software (SW) suite has been designed and developed in Matlab^®^ R2020a Programming Environment. The SW suite receives as input suitably formatted raw TWR data; it generates the corresponding correction models, performs validation checks, and finally generates validation statistics tables [8].

## 5. Position Computation Algorithm

The goal underlying this study is the development and evaluation of a suite of decentralized CP algorithms to enable the localization of multiple rovers using RF-based TWR observables collected in a network of roving and static nodes architecture. The basis of the absolute localization engine relies on Extended Kalman Filtering (EKF) realized in a collaborative manner. Considering that the adoption of a CP strategy entails the introduction of uncertainty due to the correlated positioning solutions, it is expected to affect the network solution, resulting in highly inaccurate results or even an inability for filter convergence. In an attempt to optimally combine Pedestrian-to-Pedestrian (P2P) range measurements in a decentralized manner, an approach is formulated based on Split Covariance Intersection (SCI) grounds using the inter-device TWR ranges, the advertised rover state, and covariance information.

For the localization of a mobile rover using P2I ranges, the observation setup relies on the provision of TWR observables from anchors of known coordinates to the rover in a dynamic manner. The range measurements are processed sequentially upon recording along with the reported accuracy (as estimated by the device) and the system timestamp. In a scenario of multiple rovers, each rover utilizes independently its corresponding measurements as they become available. Figure 10 illustrates the basic system setup for a single roving pedestrian and four ranges captured sequentially from the four anchors. Notwithstanding this experimental setup referring to a single rover, it may support multiple rovers subject to a potential limitation imposed by the maximum TWR technology communication network capacity.

The key elements of the EKF are the state vector (*X*), covariance (*P*), system model or state transition function (*f*), process noise (*Q*), measurement vector (*Z*), measurement covariance (*R*), measurement model (*H*), and the Kalman gain (*K*). Following typical approaches, the EKF is implemented as follows:


*Initialization*

(7)
$${\hat{X}}_{0}=\left[X_{0}\right]$$


(8)
$$P_{0}≅Q$$


*Prediction*

(9)
$${\hat{X}}_{\bar{t}}=F{\hat{X}}_{t-1}$$


(10)
$$P_{\bar{t}}=FP_{t-1}F^{T}+Q$$


*Correction*

(11)
$$K_{t}=P_{\bar{t}}H^{T}{\left(HP_{\bar{t}}H^{T}+R\right)}^{-1}$$


(12)
$${\hat{X}}_{t}={\hat{X}}_{\bar{t}}+K_{t}\left(Z_{t}-H{\hat{X}}_{\bar{t}}\right)$$


(13)
$$P_{t}=\left(I-K_{t}H\right)P_{\bar{t}}$$



### 5.1. Correction Models Adopted for the UWB and Wi-Fi RTT Range Observables Internal Accuracy

The TWR technologies employed in this study (i.e., Time Domain^©^ P410 UWB, Compulab^©^ WILD Wi-Fi RTT) result in range measurements with device-generated error values. In this study, two schemes for providing Dynamic Measurement Error Estimation (DME) algorithms are adopted and presented.

Figure 11 depicts in red the range differences obtained between the observed and reference values for a UWB rover module using a preliminary data campaign featuring four UWB anchors. Also, the same plots show in blue the median values (Leading-Edge Detection, LED) and their associated standard deviations as recorded by the sensors. According to the manufacturer, the LED flag value for LoS conditions should equal eight, whilst larger values indicate NLOS operation. Moreover, from the same plots, a relationship between the recorded LED values and the corresponding range deviations is observed, indicating that the reported values can be utilized as an index for characterizing the range quality together with their reported range error values.

In this regard, the noise of range measurements is defined by the error value reported by the UWB module for each measurement. Thorough examination of the reported (by the sensor) range errors against their estimated equivalents (computed standard deviation) suggests the introduction of a scaling factor to the reported error. Based on the relationship between the reported LED values against the computed range accuracy, an empirical scaling tactic is engaged during real-time ranging as described by the following equation:(14)$$\sigma_{t}^{rU}=\left\{\begin{array}{cc} err_{t}^{r}*5, & 7<fl_{t}^{LED}<10\\ err_{t}^{r}*10, & 10<fl_{t}^{LED} \end{array}\right.$$
where $\sigma_{t}^{rU}$ is the UWB measurement noise implemented for timestamp *t*, ${err}_{t}^{r}$ is the range error reported for timestamp *t* and fltLED is the LED flag value reported for timestamp *t*.

Preliminary examination of the relationship between the RSS values logged for the Compulab^©^ WILD units against the estimated ranging trueness values indicates the existence of a correlation. Trueness is defined as the quality metric that describes the proximity of the positioning solution in relation to its true (nominal) value. Moreover, further investigation reveals the discrepancy between the reported standard deviation (SD) values (as provided by the Wi-Fi RTT module) and the TWR measurement trueness with many instances of either overoptimistic or pessimistic SD values leading to low range quality indicator integrity. Further analysis indicates the relationship between the observed range quality of the collected Wi-Fi RTT datasets and the collected RSS values as illustrated in Figure 12. Here, the estimated range trueness scatter presents an increased distribution as the RSS values decrease, suggesting a corresponding trend.

This correlation trend is analyzed further, translating to a linear approximation of the standard deviation of range trueness against the RSS values, leading to the diagrams in Figure 13 and the following equation. This represents the measurement noise adopted for the Wi-Fi RTT observables denoted as
(15)$$\sigma_{t}^{rW}=aRSS_{t}+b$$
where $\sigma_{t}^{rW}$ is the Wi-Fi RTT measurement noise implemented for timestamp t, RSSt is the reported Receiver Signal Strength value at timestamp t, and a and b are the parameters of the linear fit model estimated empirically. This linear optimal fit is introduced during real-time ranging for dynamically assigning the range error substituting the device-generated values.

### 5.2. Kalman Filter Formulation for Distributed CP

In a similar manner to the usual EKF formulation applied for standalone positioning, the distributed collaborative positioning scheme encompassing multiple roving pedestrians relies also on the sequential processing of the recorded TWR ranges. Moreover, in addition to the P2I observables realized via the Wi-Fi RTT sensors, in this setup, the rovers are capable of performing P2P ranging operations using the UWB technology while communicating their corresponding state estimate along with their covariance matrix and utilizing them by implementing a SCIF scheme in a distributed architecture. Also, each rover is capable of storing the last available dependent and independent covariance matrices for all corresponding neighboring nodes. Figure 14 illustrates this setup for the case of two roving nodes and four anchors. Notably, this simplistic setup can be expanded to incorporate more rovers and additional anchor nodes. Obviously, in this case, a potential limitation of the maximum number of nodes/TWR observed depends on the network communication capacity. The collaborative strategy based on sequential TWR observables is formulated in a manner that could support partial or complete anchor unavailability for a certain time window throughout the localization process. As the filter state prediction and update steps rely on discrete pairwise, range-only measurements and not on range packets from multiple anchors and/or rovers, the filter is able to continuously provide a position solution for a reduced number of available neighbors (anchors or rovers). In the case of long-time windows of anchor unavailability, the filter is expected to diverge. The aim is to provide a positioning scheme robust enough to handle low P2I observable availability and extended times of P2I measurement inactivity. This is attempted through the introduction of a SCIF operation in the positioning strategy, aiming at minimizing the effect of correlation-induced errors between collaborating roving nodes.

In the proposed approach, the measurement model relies on TWR-based range-only observables enhanced by the neighbors’ state vectors and corresponding covariance matrices for providing insight regarding their position accuracy. For the case of P2I observables, the uncertainty of the anchor coordinates may be considered equal to zero; and therefore, the EKF-based absolute positioning approach may be implemented without accounting for the anchor position induced error. On the other hand, concerning the P2P observables, this approach needs to account for the moving neighbors’ position uncertainty. This is because it directly affects the filter estimation. Considering that for the P2P case, previous ranges are generated when the neighbor pedestrian is at a different position, they are, however, employed for consequent relative position estimations, which entails the fact that current state estimations between neighbors are correlated. Therefore, the neighbor’s state covariance forms the dependent part of the measurement covariance, whereas the P2P range measurement noise forms the independent part of the measurement covariance since the successive range measurements are uncorrelated.

Following the formulation of EKF, the SCIF formulation for the two estimates $\left({\hat{X}}_{\bar{t}},P_{1d}\;+\;P_{1i}\right)$ and $\left(Z_{t},P_{2d}\;+\;P_{2i}\right)$ to be combined is given by (Equation 16) through (Equation 22), where *${\hat{X}}_{\bar{t}}$* represents the state of the target node, $Z_{t}$ the TWR observable, *$P_{1d}$, $P_{2d}$* the dependent covariance matrix of the state describing the correlation between estimates, and *$P_{1i}$,$P_{2i}$* the independent covariance matrix of the state without correlation between estimates. The resulting state estimate is denoted by *(${\hat{X}}_{t},\;P_{d}+P_{i}$)* with its associated covariance matrix described by dependent and independent parts, accordingly:(16)$$P_{1}=\frac{P_{1d}}{\omega }+P_{1i}$$
(17)$$P_{2}=\frac{P_{2d}}{1-\omega }+P_{2i}$$
(18)$$K=P_{1}H^{T}{\left(HP_{1}H^{T}+HP_{2}H^{T}\right)}^{-1}$$
(19)$${\hat{X}}_{t}={\hat{X}}_{\bar{t}}+K\left(Z_{t}-H{\hat{X}}_{\bar{t}}\right)$$
(20)$$P=\left(I-KH\right)P_{1}$$
(21)$$P_{i}=\left(I-KH\right)P_{1i}{\left(I-KH\right)}^{T}+KP_{2i}K^{T}$$
(22)$$P_{d}=P-P_{i}$$
where the ω∈ [0, 1] coefficient is selected, subject to minimizing the determinant of the resulting fused covariance matrix [50].

Figure 15 presents the flow chart of the developed distributed collaborative positioning (DCP) algorithm, summarizing its functionality.

## 6. Data Collection and Error Mitigation

Section 6 aims at introducing and discussing the procedures adopted for the generation and collection of simulated and field range data, respectively, for testing the proposed positioning algorithms. Also, it presents the experimental evaluation procedures and techniques used for error mitigation.

### 6.1. Test Data Summary and Equipment Employed

The experimental campaigns include data collection undertaken both outdoors and indoors. Outdoor campaigns serve as early-stage feedback of the performance of TWR technologies examined in this work, while at the same time providing a basis for the planning of the indoor experiments. Indoor campaigns serve both as a means for the detailed examination of the range error mitigation models in challenging conditions, as well as for the development and evaluation of the kinematic position technique developed. Performance assessment of the range correction models is implemented both for the UWB and Wi-Fi RTT sensors on static as well as kinematic data. Finally, testing with simulated datasets is crucial, as it enables the generation of controlled and realistic TWR datasets in a systematic manner, facilitating the development and optimization of the proposed CP algorithms.

The UWB system employed for field testing in some of the campaigns is the P410 module by Time Domain^®^. Its principle of operation relies on the coherent transmission of very-short-duration RF waveforms. The high resolution of the transmitted RF pulses offers the ability to perform high-accuracy range measurements, including capabilities of identifying and rejecting NLOS and multipath ranges. The nominal high range accuracy of the P410 module reported by the manufacturer relies on the ability of the transceivers to precisely identify the first received pulse known also as the Leading-Edge Detection (LED) feature.

For Wi-Fi RTT, the Compulab^®^ Wi-Fi Indoor Location Device (WILD) modules are utilized. They are among the first commercially available devices that support the communication with FTM (Fine Time Measurement)-compatible Android smartphones. The successful FTM ranging relies on the support of Wi-Fi RTT API by the smartphone and through dedicated Android applications. The operation of WILD units relies on the Compulab fitlet2 platform that encompasses an Intel AC8260 Wi-Fi processor unit.

The employed Android smartphone devices are both manufactured by Google, as they were the first commercially available devices to support the IEEE 802.11mc FTM protocol [54]. For one campaign, the Wi-Fi RTT observables are collected using a Google Pixel 2 device employing a Qualcomm^®^ MSM8998 Snapdragon 835 chipset. The Android smartphone utilized in the second campaign is the Google Pixel 3a XLTM device that supports the FTM protocol, enabling Wi-Fi RTT ranging. This device also enables the collection of azimuth values, utilizing the embedded inertial (accelerometer, gyroscope, and magnetometer) sensors. During data collection, the Android 9 software was installed on both smartphones.

### 6.2. Indoor Field Test Campaigns

One field test campaign aimed at the examination of UWB observables both in terms of range correction as well as trajectory determination. The test took place indoors within the premises of SRSGE, NTUA. The laboratory area included two separate office areas connected with a small corridor and a third smaller room offering the ability to collect UWB ranges both in LOS and NLOS conditions. Concerning range correction assessment, a number of correction and validation points were defined in order to cover the entire area in a uniform manner. Specifically, five correction points were established in two rooms and one correction point in a third room. Similarly, three validation points were established in the first two rooms and two in Room 3 [8].

Another field test campaign examined WiFi-RTT both for assessing range correction models as well as for testing the pedestrian kinematic positioning algorithms. Field testing took place at the lobby and corridor area located within the “Lampadario” building of SRSGE, NTUA. The effective area included a portion of the corridor which spanned approximately 20 m in length and 3 m in width as well as the adjacent lobby with an area of around 70 m^2^, providing a total area of around 125 m^2^.

### 6.3. Range Errors Mitigation

#### 6.3.1. UWB Indoor Range Correction Analysis

For the performance analyses of the range correction models development, a rover–anchor TWR dataset was collected for each correction point. As an example, the histograms in Figure 16 depict the probability density function of the range dataset collected for every pair of UWB nodes at correction point C1. The Freedman–Diaconis rule is used for optimizing the bin size selection. The Empirical Probability Density Function is estimated using kernel density estimation with a kernel bandwidth value of 0.005, resulting in a good fit for the P410 module UWB data. Conclusively, the ranging values used for further processing are the ones with the highest probability density (EPDFmax) as the most representative of the samples. The necessity of a range correction technique is obvious based on the offset with respect to the corresponding reference distance value.

In Figure 17, the bi-dimensional correction fit is presented, which relies on the location of each correction point using its correction value. In order to cover the entire test area, the correction values are interpolated using natural neighbor interpolation, which is based on the Voronoi tessellation method (see Section 4.2.2), and therefore, this correction approach is referred to as “Voronoi correction” or “vc”. For the area found outside the polygons defined by the correction points, a linear extrapolation is performed in order to extend the Voronoi correction values.

This approach is expected to offer the most effective correction solution by capturing the fluctuations in correction values resulting from environmental factors. These factors can arise from changes in the inter-node distance or the impact of the Non-Line of Sight (NLOS) ranging through different materials. Figure 17 shows the results obtained from the “vc” method. The models indicate an apparent increase in the correction values for the NLOS areas. Another remark that relates to the values generated by the spatial extrapolation, and especially for the right-most area of Room 1 (top) for pair 300-302, is an irregular behavior that most likely relates to the extreme values found in the right-most area of Room 2 and the slight offset (towards right) of C7 and C10 with respect to locations of points C2 and C5.

For the static ranges validation, the range measurements collected on the validation points are exploited in two stages in order to evaluate the efficiency of the three correction models: firstly, before applying any range correction and after correction values have been implemented. Figure 18 shows the results obtained for point V1 in the form of histograms along with the generated EPDFmax values for each correction model. In the same plots, the reference distance (in yellow vertical lines) illustrates the improvement in comparison with the uncorrected values.

The diagram presented in Figure 19 summarizes the performance of all validation points for each correction. It reports the mean deviation from the reference distance and its standard deviation value for all UWB pairs. As expected, all correction models result in improved solutions compared to the “NoCorr” results. Moreover, differences in the performance between methods are recognized. In summary, the “arlc” technique offers less improved results, whereas the performance of “vc” proves to be marginally better compared to “rlc”. Overall, the improvement compared to the “NoCorr” results ranges from 32 to 86%.

#### 6.3.2. Wi-Fi RTT Indoor Range Correction Analysis

Using a similar procedure to the one employed for UWB positioning in campaign 1, the TWR observables collected between the rover and all anchor APs are processed to estimate the statistics and associated correction values. Figure 20 presents the range observables between the rover and anchors for correction point C1 in the south orientation (C1s). Again, the EPDFmax value is estimated, for which the kernel bandwidth value 0.02 is adopted to optimally fit the data.

From Figure 20, it becomes evident that there is a necessity for the development and implementation of a range correction model with cases of range bias values of up to 8 m. The correction models that are developed rely on the “arlc” and “vc” models expanded suitably to incorporate the orientation parameter. The resultant models are the “orientation–linear correction” (“olc”) and the “orientation–Voronoi correction” (“ovc”) models (see Section 4.3). As an example, Figure 21 presents the “olc” models at the south and north orientations for the 901-301 rover–anchor pair; the apparent variation between the models indicates the necessity for further examining the distinct orientation models effect.

Figure 22 presents the results for the “ovc” model for 901-301 rover–anchor pair at south and north orientations. Again, the apparent variation between the different orientation models indicates the necessity of examining the effect of the oriented models. Moreover, the effect of adjacent walls is apparent by the resulting range correction patterns. Interestingly, from these plots, it is possible to identify whether the respective anchor is located within a corridor due to the elongated pattern of the range correction values. The variability observed in the “olc” and “ovc” values for each anchor strongly implies that the environmental conditions surrounding the anchor locations play a significant role in the subsequent model generation.

Using the three receptive correction models for the range datasets collected at the validation points, the model evaluation is implemented. Figure 23 presents the histograms generated in the south orientation of validation point 2 (V2s) after the correction models are applied. Both “lc” and “vc” models present an initial improvement in the resulting ranges, however, without indicating the predominance of a specific model.

The validation results for all VPs are summarized in Figure 24 for the respective correction models after combining the values of the different orientation models. The results suggest the potential of the “vc” model for producing better performance in comparison to the “lc” model. The small discrepancy in the results between the two approaches indicates the potential of both for the next analysis steps regarding the effect of the correction models using kinematic datasets.

## 7. Position Solution Estimation

This section presents the experimental results obtained for the position solution using the combined UWB/Wi-Fi RTT algorithmic approach and the data sources detailed in Section 5 and Section 6, respectively. The evaluation of the proposed positioning techniques for non-collaborative rovers relies both on simulated and field data. The evaluation of the collaborative positioning scheme, due to hardware limitations and adversities, relies only on exhaustive simulated datasets generated suitably for multiple, simultaneously operating rovers in varying availability conditions.

### 7.1. Localization Solutions Obtained Using Field Data

In this section, we present the performance evaluation of the proposed positioning algorithms using the field data obtained in indoor environments. The objective is to assess the algorithms’ resilience in addressing the challenges encountered when working with real TWR datasets affected by errors arising from hardware limitations and environmental factors.

#### 7.1.1. UWB Indoor Trajectory Computation

The positioning stage of campaign 1 relies on the ranges collected between a single rover and all available anchors. The position estimation of the mobile node is attained by employing the EKF algorithm. The noise of the range measurements adopted in the filtering process corresponds to the error values reported by the UWB module for each measurement. Specifically, based on the relation between the reported LED values and their precision, an empirical scaling tactic is engaged during trajectory estimation. Alternative correction methods are examined individually via implementing the correction values to the ranges for each EKF run and in a dynamic manner. For the case of linear corrections, the range correction value used is calculated based on the reported range value by the device, whilst for the Voronoi correction approach, the corresponding value is established based on the last known position estimated using the EKF. Figure 25 shows the results obtained. In order to facilitate comparisons, the estimated trajectories are overlaid on the reference travel path. More specifically, the plots of Figure 25 provide a graphical representation of the performance of each method. The user moves starting from the top-left corner of Room 1 and concludes at the bottom right of Room 2. The short stop-and-go sections are evident in the vicinity of each travel path, realized at the spots for which a point cluster is observed (blue dots), while the linear segments of the trajectory connect these clouds.

Thorough examination of Figure 25 reveals a number of conclusions. Considering the “NoCorr” rover trajectory, in general, it follows the actual path; however, significant deviations from the ground truth are evident. A systematic offset from the true trajectory and sections that appear to be crossing the walls are apparent due to excessive range errors. The results derived for the “rlc” correction method present a noticeable improvement compared to the raw observable solution, with a significant part of the trajectory precisely following the true travel, particularly in the section close to the check points. Regarding the trajectory solution computed using the “arlc” correction model, an improvement is also remarkable compared to the “NoCorr” method with the entire trajectory closely following the true travel path. Clearly, there exist no points crossing the wall barriers; however, some larger deviations appear compared to the “rlc” technique. Finally, the trajectory generated with the “vc” method reveals an overall improvement observed against all other correction methods. The rover trajectory is more stable and lies closer to the true path with one exception at the corridor pass from Room 1 to Room 2, where all the correction methods present a weakness. This weakness is most probably the result of missing correction points at the boundary areas, such as narrow passes between rooms with unstable RF behavior. In this occasion, the corridor area correction is produced with interpolated data, which effectively lack the necessary resolution required for boundary conditions. Overall, the implementation of any correction method improves the mobile node position solution, with the “vc” approach providing superior performance. Table 3 provides a summary of the results of the aforementioned analyses, in which the horizontal trueness is expressed in the mean value, the standard deviation, and max value for each method.

#### 7.1.2. Wi-Fi RTT Indoor Trajectory Estimation

Trajectory computation for campaign 2 relies on the ranges collected among the rover and the available Wi-Fi RTT access points. The estimation of the rover’s position is carried out using the EKF. The noise in the range observations is determined by the error value estimated for each measurement. To account for the relationship between the reported RSS values and their corresponding accuracy, an empirical scaling technique is employed during trajectory estimation. Two scenarios realized on the same path are undertaken for a pedestrian walking indoors, starting from the lobby area (right-most part of Figure 26) towards and into the corridor area (left-most part of Figure 26). The only difference between the two data collection scenarios is the walking speed; the first and second scenarios are performed at slow and standard walking paces, respectively. The slow pace scenario enables an increase in the number of logged range samples, as it associates with a sampling rate of up to 5.9 Hz. Different range correction approaches are implemented for both scenarios and evaluated against the reference path, providing a comparative assessment of their performance. Figure 26 illustrates the rover trajectories computed for the alternative correction models for S1. Apparently, if no range correction model is applied, the positioning algorithm performs poorly, with the solution rapidly diverging from the ground truth. In comparison to the “no correction” case, all range correction models perform significantly better, especially for the lobby section, where the anchor geometry is balanced. Based on the rover trajectories extracted, all the correction models conclude with results of similar quality, with the “arlc” and “olc” models offering more stable solutions with smoother transition parts.

Figure 27 summarizes the implication of range correction models on the rover position based on their ECDF graphs. The “no correction” model contributes a position trueness of worse than 5 m at 50% of the sample, whereas the correction trueness of the correction models ranges close to 2 m at 50% of the sample. The differences observed in the performance between the linear (“arlc”, “olc”) and spatial (“vc”, “ovc”) range correction models are depicted for 95% of the sample. The linear model results in a position fix trueness of 2.5 m, whereas the spatial ones result in trueness values of 4 m (“vc”) and 5.1 m (“ovc”), respectively.

The performance statistics of campaign for various range correction models are summarized in Table 4. It is evident that the corridor’s deteriorating geometric effect is observed in the trajectories. Additionally, the spatial error correction methods demonstrate limitations in effectively mitigating the adverse impact of degraded observables in a consistent manner.

### 7.2. Distributed Collaborative Positioning (DCP) Using Wi-Fi RTT P2I and UWB P2P Simulated Data

The Wi-Fi RTT/UWB fully collaborative P2I/P2P positioning simulation test trials employ four rover and four anchor nodes. The first scenario, with uninterrupted availability of the anchor nodes, serves as a baseline for the evaluation of the developed DCP algorithm in optimal conditions. The selected subsequent scenario incorporates two intentionally induced time windows (at 8 s and 30 s), designed suitably to simulate degradation in the anchor availability. Specifically, the varying length of the anchor availability windows are designed to simulate the dynamic anchor connectivity loss, typically found indoors. The DCP algorithm is therefore examined for its robustness. This is undertaken both for a short and a long data loss window. Regarding anchor availability, the trials examine different combinations of anchor loss for a number of cases spanning from one up to four anchor points (i.e., complete anchor unavailability).

#### 7.2.1. DCP P2I/P2P with No Anchor Loss

The initial scenario involves the implementation of distinct positioning algorithms that support P2I and P2I/P2P functionality. The purpose is to validate these algorithms and evaluate their positioning performance using an ideal dataset that does not experience any communication loss. Figure 28 demonstrates the capability of both approaches to estimate trajectories that closely align with the reference trajectories. However, the impact of noisy Wi-Fi RTT observables becomes evident, as the trajectories of the respective approaches (EKF and EKF/Az) exhibit outlier events and less smooth positioning solutions. Conversely, the CPKF and CPKF/Az solutions show significant improvements in positioning, characterized by smoother trajectories and closer alignment with the reference solution. The results obtained and depicted in Figure 29 confirm the positioning performance potential of both approaches. It is noted that DOP and DOP-vAnc provide the same exact values for no anchor loss. Although the CP solutions exhibit smoother characteristics as shown in the third row of the figure, the corresponding Empirical Cumulative Distribution Function (ECDF) plots demonstrate that the computationally and communicationally less demanding KF implementations are capable of functioning adequately. This suggests that the adoption of CP could prove redundant under some conditions (i.e., in cases of a fully operational anchor network), as standalone KF algorithm implementations can fulfill the minimum positioning requirements effectively.

#### 7.2.2. DCP P2I/P2P for Complete Anchor Loss

For the “complete-anchor-loss” scenario, the anchor loss corresponds to all four anchors, resulting in a total data loss for 16 node pairs during anchor unavailability events. As illustrated in Figure 30, the ”KF”, “KF/Az”, and “CPKF” positioning solutions once again exhibit extreme position errors, rendering them unable to provide accurate position fixes during unavailability events. However, the ”CPKF/Az” solution demonstrates its capability to closely align with the reference positions, maintaining a satisfactory level of performance. As illustrated in Figure 30, the ”KF”, “KF/Az” and “CPKF” positioning solutions once again exhibit extreme position errors, rendering them unable to provide accurate position fixes during unavailability events. However, the ”CPKF/Az” solution demonstrates its capability to closely align with the reference positions, maintaining a satisfactory level of performance. As illustrated in Figure 30, the ”KF”, “KF/Az”, and “CPKF” positioning solutions once again exhibit extreme position errors, rendering them unable to provide accurate position fixes during unavailability events. However, the ”CPKF/Az” solution demonstrates its capability to closely align with the reference positions, maintaining a satisfactory level of performance. Figure 31 emphasizes the potential of the proposed “CPKF/Az” approach. It showcases maximum position trueness values of approximately 4 m during periods of unavailability, despite the extreme values observed in the “DOP CP” metric. This highlights the robustness of the approach in handling such highly challenging conditions.

#### 7.2.3. Discussion of Performance for the Different Positioning Algorithms

A comprehensive analysis and summary of the achieved positioning results show the following results. Figure 32 summarizes the results obtained for positioning the trueness for the trajectories of the different simulation-based campaigns, indicating their strengths and weaknesses and providing insight regarding the potential of the proposed algorithms. Apparently, the introduction of UWB combination with the Wi-Fi RTT observables in a realistic configuration (i.e., Wi-Fi RTT for P2I and UWB for P2P) enhances the resulting solution. Azimuth observables further improve the positioning results, increasing the system’s robustness and efficiency since they contribute to obtaining consistently accurate and smooth solutions of high availability. The introduction of a single UWB anchor in a P2I configuration offers 35.1% improvement in position trueness. Moreover, the inclusion of azimuth observables results in 15.7% improvement in position trueness for the Wi-Fi RTT-only solutions, while it provides similar enhancement for all campaigns. The highlight of the azimuth effect is apparent in the “all anchor loss” scenario of 3. In this case, it enables trueness improvement of 38.1% for the standalone solution and 85.1% trueness improvement for the P2I-P2P solution. This observation underlines the necessity of orientation information for the successful implementation of the Covariance Intersection Filter in order for the solution to converge. Regarding the position availability, we observe values of 100% even for time windows featuring one available anchor and for the standalone (P2I) approach. Such behavior is indicative of the effect of the proposed approach design that relies on sequential ranging utilization. This is indicative of its robustness, in contrast with traditional trilateration-based approaches that require the collection sets of ranges (minimum 3) prior position estimation. In the case of complete data loss (i.e., “all anchor loss”), the proposed DCP algorithm operates successfully, providing a positioning solution of stable quality regarding the reported trueness, as well as 100% availability. In contrast, P2I-only approaches offer up to 74.3% availability, coinciding with the complete anchor loss time windows, which account for approximately 25% of the total trajectory time. With that being said, it is important to acknowledge that the reported 100% availability measure should be viewed as overly optimistic. This could be due to the completely controllable simulated conditions, which fail to account for potential data loss events caused by device malfunctions and hardware limitations, which may arise in real-world scenarios.

## 8. Conclusions and Outlook on Future Work

The proposed approach stands out due to its originality in two main aspects. Firstly, it focuses on developing and evaluating suitable models for correcting range errors in RF-based TWR technologies. Secondly, it emphasizes the development of a robust CP engine for groups of pedestrians. This engine is designed to handle scenarios with limited anchor availability while also ensuring scalability through a distributed positioning architecture using a single-level setup of collaborating nodes (i.e., all nodes are identical, and no primary/secondary classification is required). The impact regarding the development and systematic evaluation of empirical range error correction models for UWB and Wi-Fi RTT is summarized in the following:Development and implementation of spatial (2D) error corrections models for RF-based technologies.Introduction of orientation and RSS information within the corrections models.Detailed and systematic performance evaluation of the proposed correction models leading to corresponding variations in both UWB and Wi-Fi RTT technologies.

In addition, the originality regarding the development and implementation of the pedestrian indoor CP algorithm refers to the following:The combined use of Wi-Fi RTT and UWB in order to provide a balanced solution by utilizing the strengths and restrictions of each technology correspondingly;The ability of the algorithm to operate efficiently while a minimum number of anchor nodes is available for short periods by optimally combining P2P range measurements;The utilization of a range/heading Split Covariance Intersection Filter for UWB/Wi-Fi RTT/IMU loosely coupled fusion in order to provide robust indoor positioning for groups of pedestrians.

Overall, the assessment of the proposed methodologies reveals an improvement in position trueness for UWB and Wi-Fi RTT cases of the orders of 74% and 54%, respectively. The proposed localization algorithm based on a P2I/P2P configuration provides a potential improvement in the position trueness of up to 10% for continuous anchor availability. Its full potential is evident for short-duration events of complete anchor loss (P2P-only), where an improvement of up to 53% in position trueness is achieved. Overall, the performance metrics estimated based on the extensive evaluation campaigns demonstrate the effectiveness of the proposed methodologies. The identified limitations of this work can be summarized in the following aspects.

The assessment of the range error correction models in limited LoS/NLoS conditions, undermining the ability to generalize the results;The functionalities of the range observables simulator do not include, in the current state, abilities to dynamically replicate varying signal transmission conditions (i.e., NLoS and multipath);The testing of the DCP algorithm needs to be further evaluated in real-world scenarios;The assessment of the potential channel congestion for scenarios of multiple users in real world scenarios needs to be considered.

Further enhancements in the system, as well as the ability to further investigate the different variations in the proposed approaches, enable future expansion. Potential future work and scope include the following:Implementation and assessment of the range error correction models in varying environments. Typical environments have been utilized in order to (1) analyze the impact of the environmental effects pertaining to specific area types, and (2) evaluate the validity of the adopted and proposed range error mitigation approaches. Evaluating the correction methodology in different test areas with varying LOS/NLOS conditions can further support its generalization ability. Moreover, extensive datasets can be utilized for potentially introducing data-driven AI techniques (i.e., machine learning) for investigating the ability to (1) minimize the required number of check points, and (2) to select the optimal checkpoints’ locations based on multiple parameters (i.e., building geometry, TWR technology specifications, and maximum field campaign duration).As the proposed range error evaluation approach can be expanded to a virtually unlimited number of similar technologies, further evaluating the developed software with additional RF-based ranging datasets (i.e., low-cost UWB sensors) is suggested. By performing experimental campaigns on the same test areas, baseline comparisons may be provided; subsequently, further configuration and fine-tuning of the methodology would be enabled, facilitating future methodology generalization.An extension of the range observables simulator, including the ability to simulate NLOS ranges through varying materials, would further enhance its robustness. The configuration of the simulation variables could rely on existing through-the-wall RF transmission models and additional field campaigns for calibrating them with additional datasets. Moreover, multipath-generated ranges could be introduced through, for example, suitable combinations of ray-tracing techniques and Monte Carlo methods. This would enhance the ability of the simulator to provide realistic datasets, facilitating future investigations of complex multitechnology, multi-environment scenarios.As the DCP algorithm is designed based on the distributed collaborative architecture, offering scalability and the ability to facilitate future implementation on mobile devices, it is suitable for a number of relevant applications. Notwithstanding a great number of personal mobility applications relying directly on the positioning solution produced using a single device (i.e., smartphone), a continuously increasing number of them rely on additional state information (orientation, elevation, etc.). Given the multisensory character of today’s smartphones, a great number of applications could benefit from the fusion of additional sensor data introduced within the loosely coupled architecture of the DCP solution. For example, as the UWB functionality is already available for a number of smartphones and given the cost limitations implicated by these mass-market devices, the investigation of the proposed approaches using low-cost UWB sensors would provide valuable insight regarding their large-scale applicability. Moreover, the provision of elevation information through barometric sensors data, or the inclusion of indoor maps that would set boundaries for the kinematic trajectory (i.e., map-matching approaches), would potentially increase the solution robustness. Both the improvement in the rover self-localization, as well as the consequent collaborative steps that would propagate the quality improvement to neighbor nodes, would benefit a potentially unlimited number of users.

## Figures and Tables

**Figure 1 sensors-24-07520-f001:**
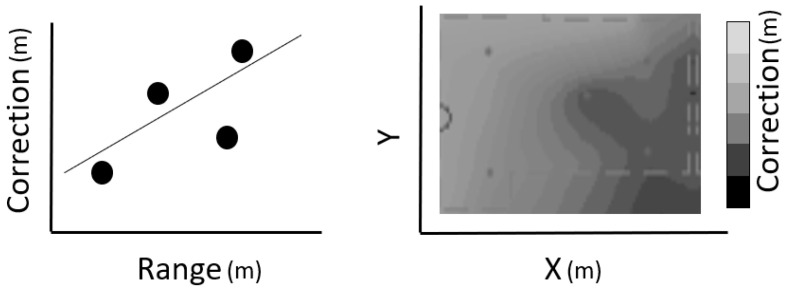
Empirical (spatial) error correction models: 1D model (**left**), 2D model (**right**).

**Figure 2 sensors-24-07520-f002:**
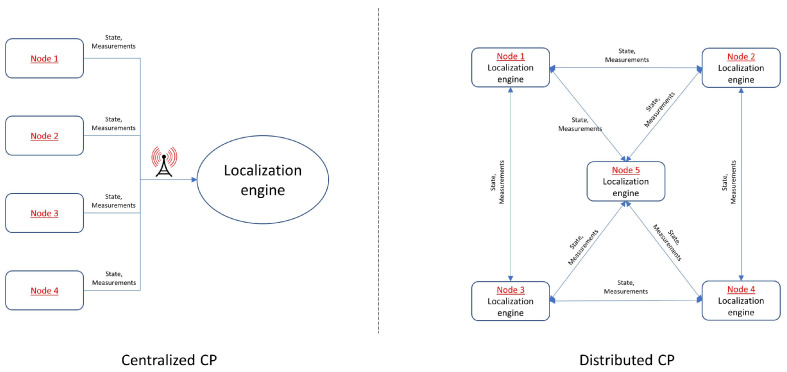
Distinction between centralized CP architecture (**left**) and distributed CP architecture (**right**).

**Figure 3 sensors-24-07520-f003:**
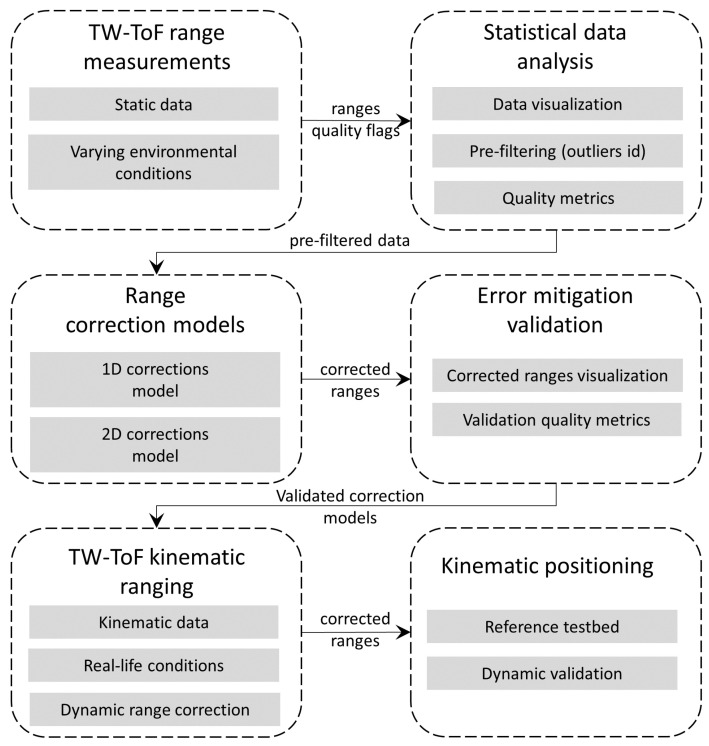
TWR range correction methodology steps.

**Figure 4 sensors-24-07520-f004:**
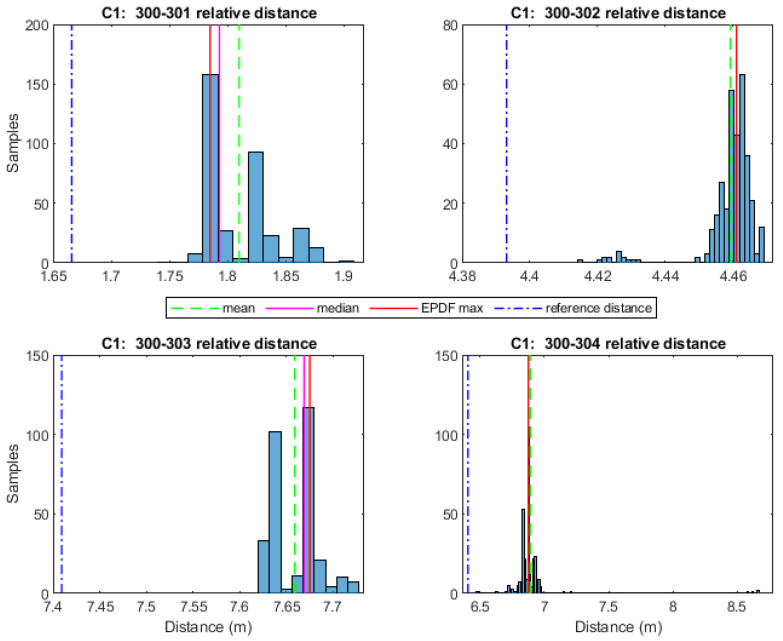
UWB P410 ranges histograms and representative statistical values.

**Figure 5 sensors-24-07520-f005:**
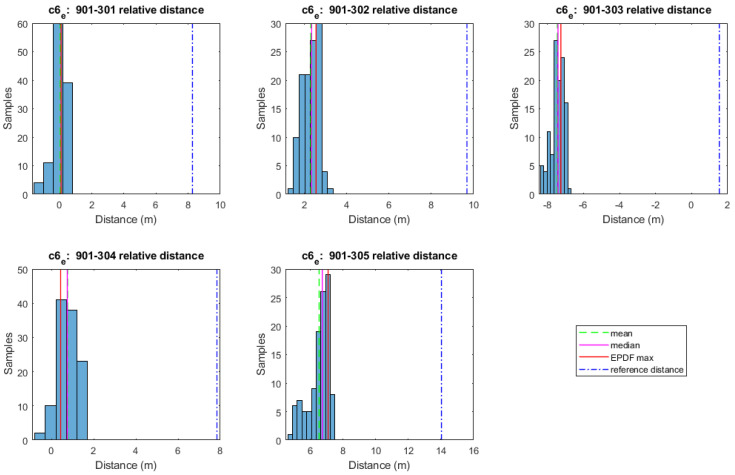
Wi-Fi RTT WILD ranges histograms and representative statistical values.

**Figure 6 sensors-24-07520-f006:**
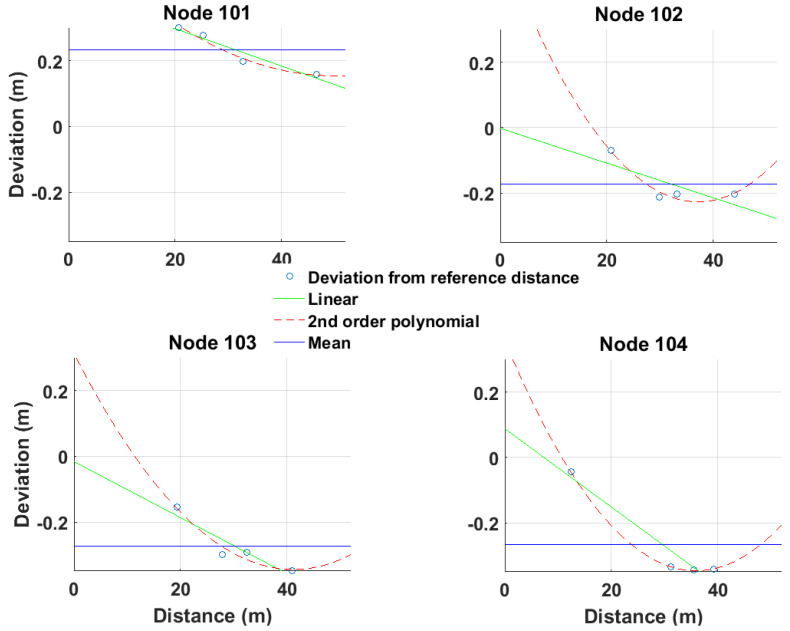
Example radial (1D) range correction models for UWB (P410 Time Domain^©^) data.

**Figure 7 sensors-24-07520-f007:**
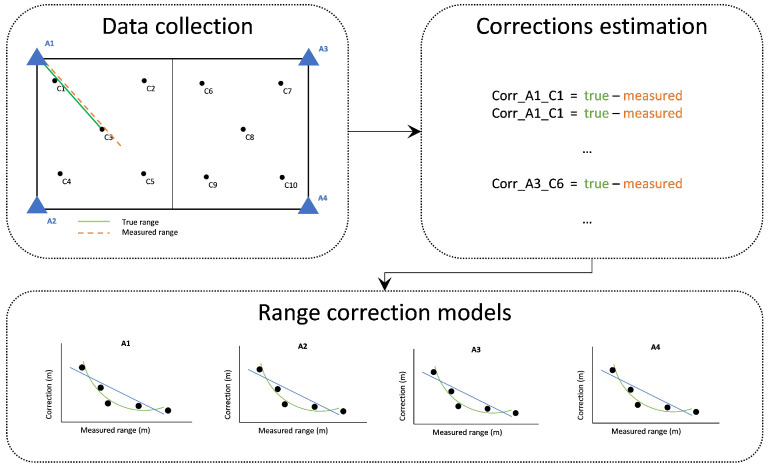
Empirical 1D range correction models estimation.

**Figure 8 sensors-24-07520-f008:**
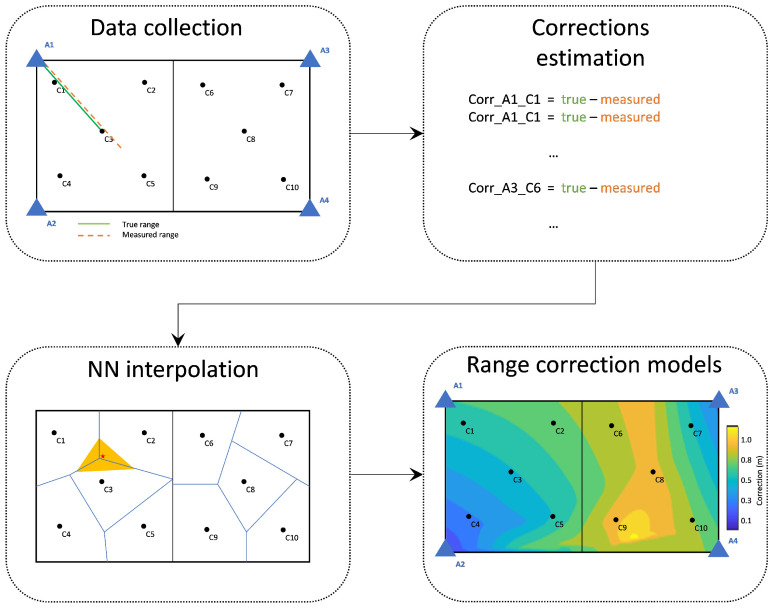
Empirical 2D range correction models estimation.

**Figure 9 sensors-24-07520-f009:**
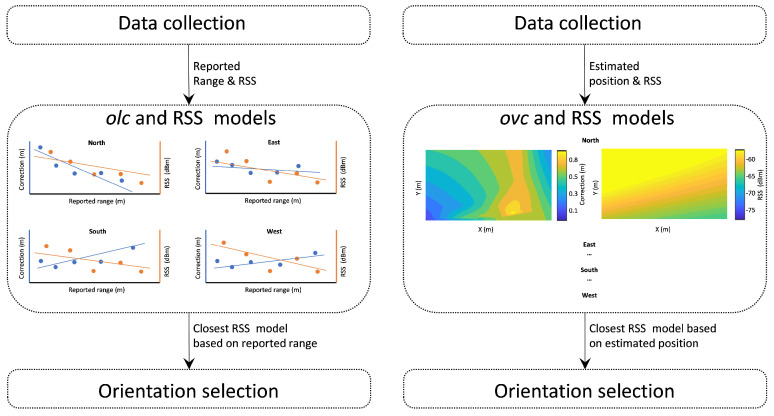
Proposed RSS-based orientation selection approaches. Radial-based selection (**left**) and bi-dimensional-based selection (**right**).

**Figure 10 sensors-24-07520-f010:**
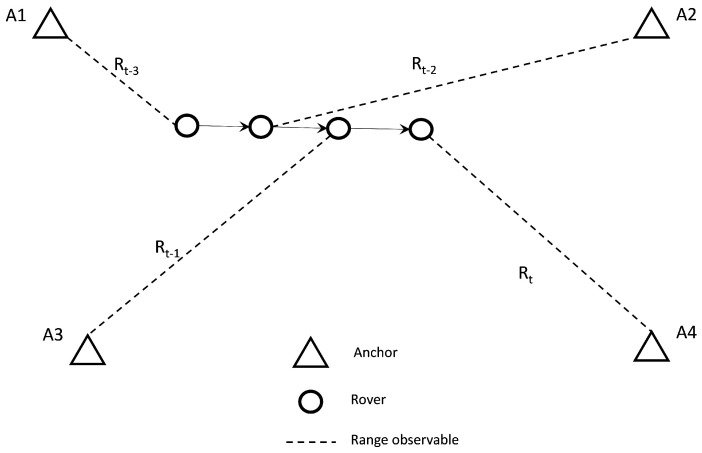
TWR ranging setup for a single rover EKF-based localization.

**Figure 11 sensors-24-07520-f011:**
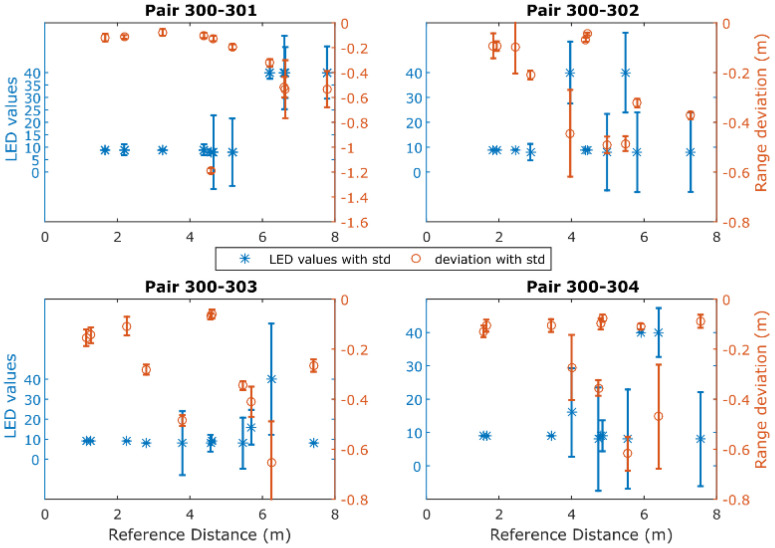
LED (Leading−Edge Detection) flags with corresponding range deviations along with the standard deviation values for all UWB pairs.

**Figure 12 sensors-24-07520-f012:**
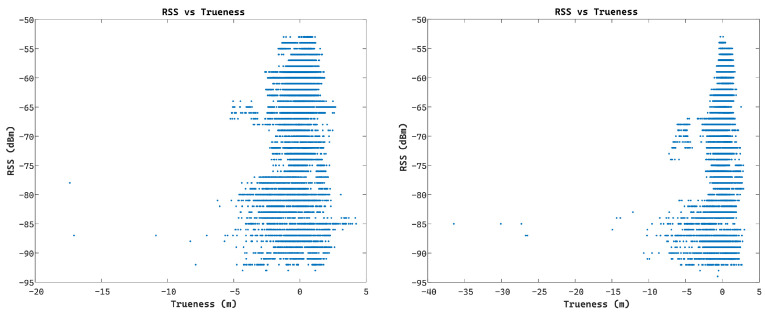
Empirical RSS versus trueness diagrams for Wi-Fi RTT observables.

**Figure 13 sensors-24-07520-f013:**
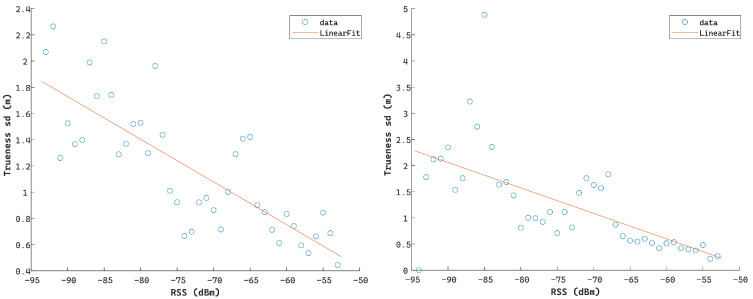
Examples of empirical trueness SD versus RSS values for Wi-Fi RTT observables.

**Figure 14 sensors-24-07520-f014:**
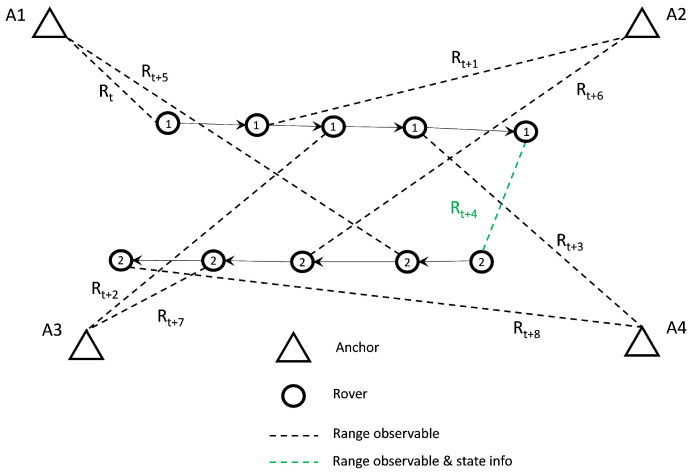
TWR ranging and communication setup for two rovers’ SCIF-based localization.

**Figure 15 sensors-24-07520-f015:**
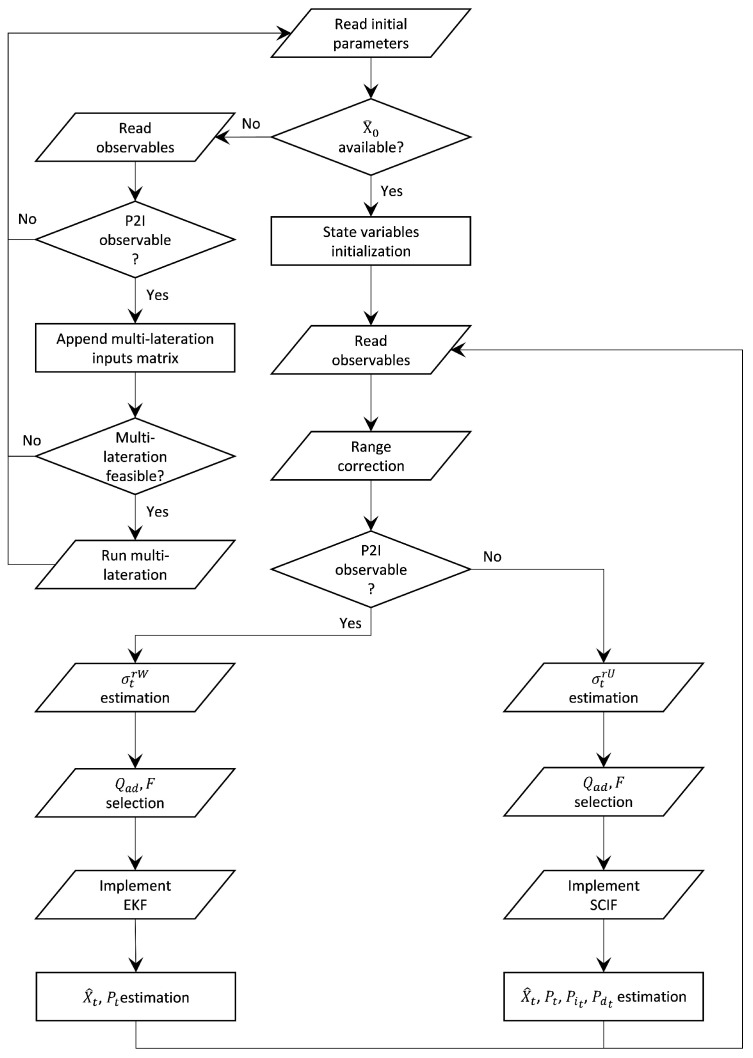
DCP (distributed collaborative positioning) algorithm implementation diagram, illustrating the respective data flows, error correction implementation, and adaptive filtering steps, as well as standalone or collaborative positioning.

**Figure 16 sensors-24-07520-f016:**
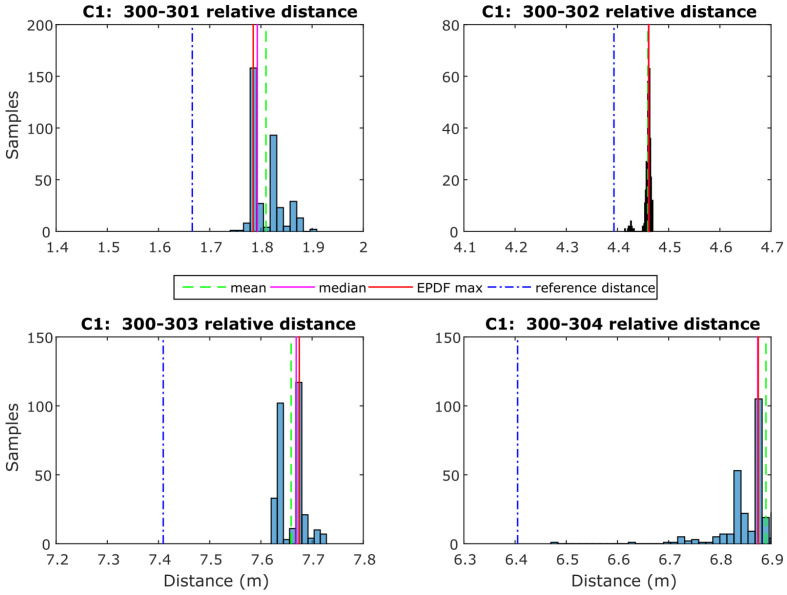
Range histograms for all UWB node pairs at point C1 for campaign 1.

**Figure 17 sensors-24-07520-f017:**
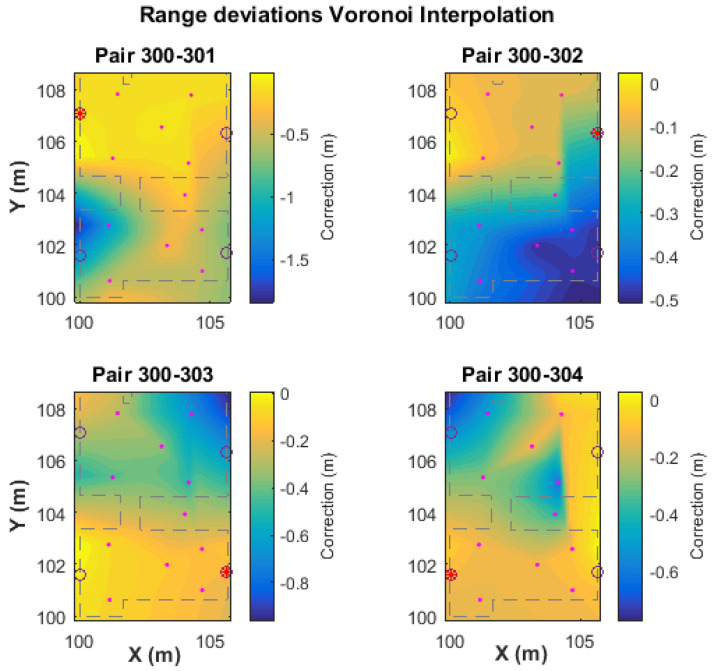
Bi-dimensional interpolated range error Voronoi surfaces for the different UWB pairs for campaign 1.

**Figure 18 sensors-24-07520-f018:**
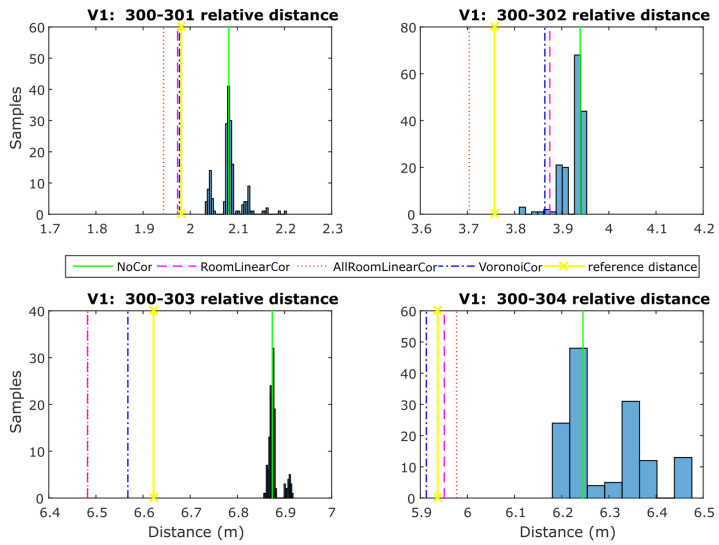
UWB ranges histograms along with calibrated “EPDFmax” values for the different correction methods at point V1 for campaign 1.

**Figure 19 sensors-24-07520-f019:**
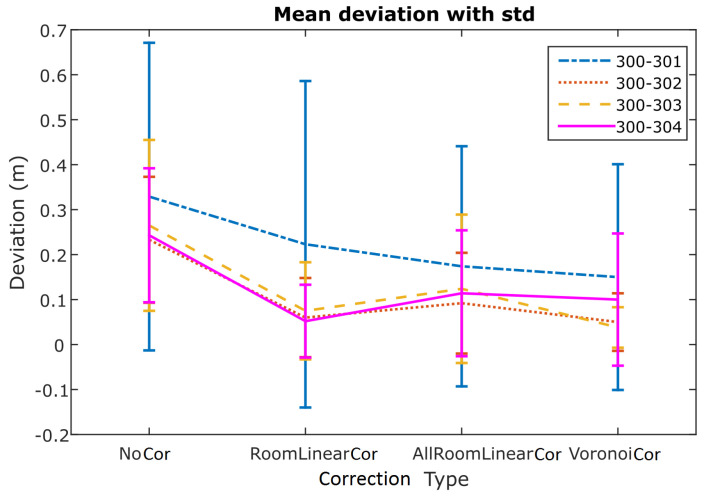
UWB ranging mean trueness with standard deviation values per correction method using all validation points for campaign 1.

**Figure 20 sensors-24-07520-f020:**
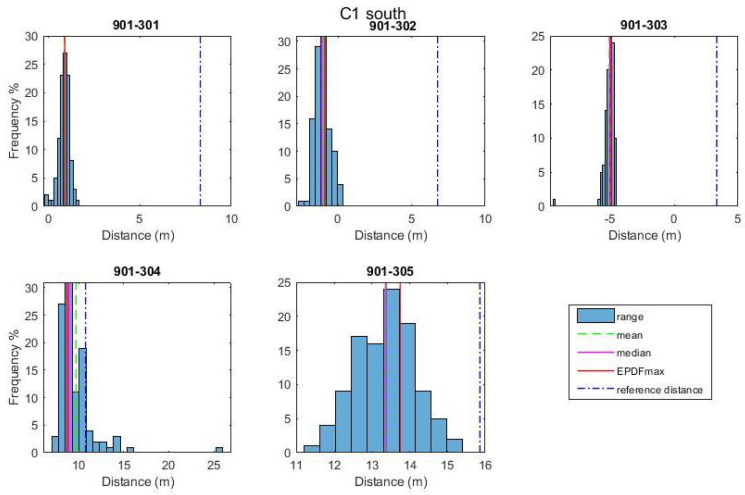
Range histograms for all Wi-Fi RTT APs at point C1_south for campaign 2.

**Figure 21 sensors-24-07520-f021:**
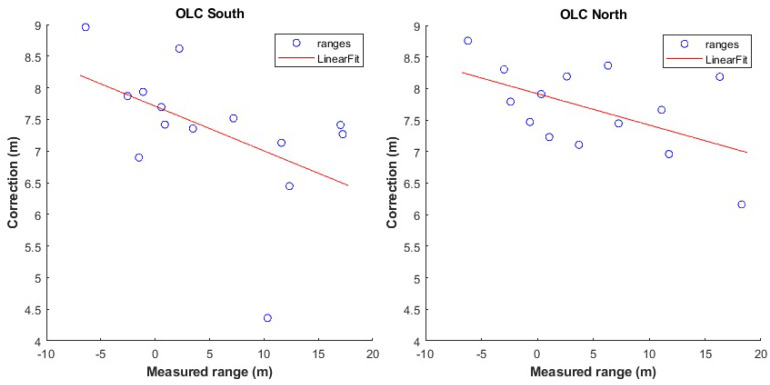
Correction models for south and north orientation–linear correction (OLC) estimated for the 901-301 Wi-Fi RTT pair of campaign 2.

**Figure 22 sensors-24-07520-f022:**
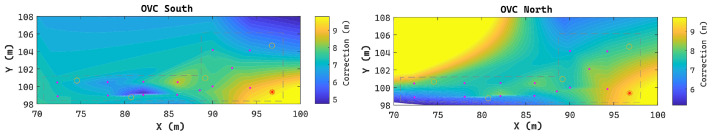
Bi-dimensional interpolated south and north orientation–Voronoi correction (OVC) range error Voronoi surfaces for the 901-301 Wi-Fi RTT pair for campaign 2.

**Figure 23 sensors-24-07520-f023:**
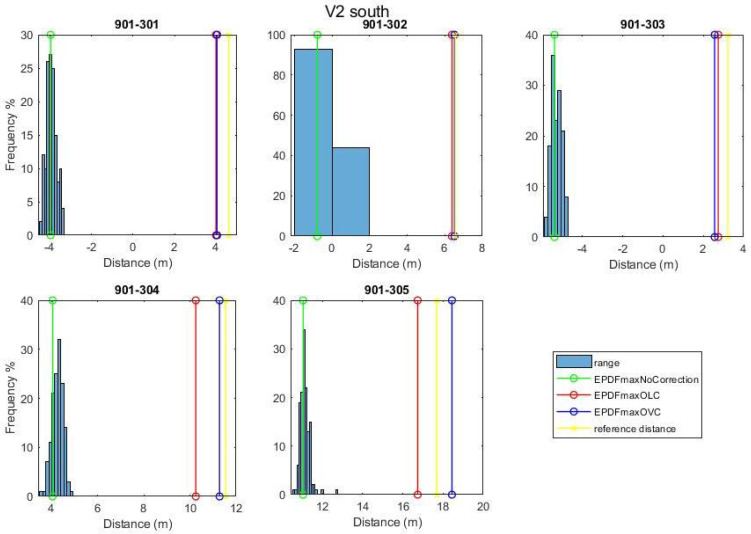
Wi-Fi RTT range histograms along with calibrated “EPDFmax” values for the different correction methods at point V2 for campaign 2.

**Figure 24 sensors-24-07520-f024:**
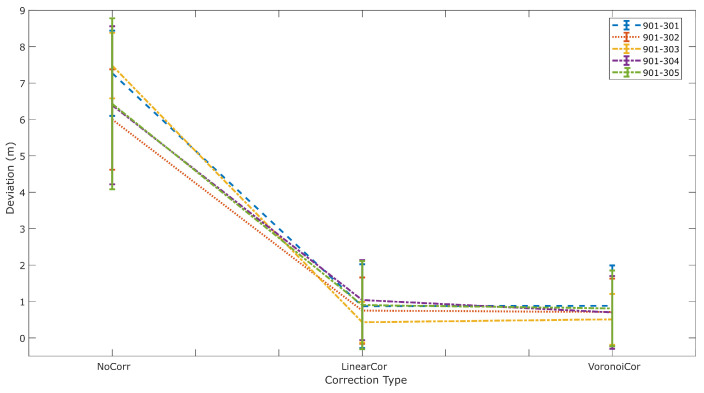
Wi-Fi RTT ranging mean trueness with standard deviation values per correction method using all validation points for campaign 2.

**Figure 25 sensors-24-07520-f025:**
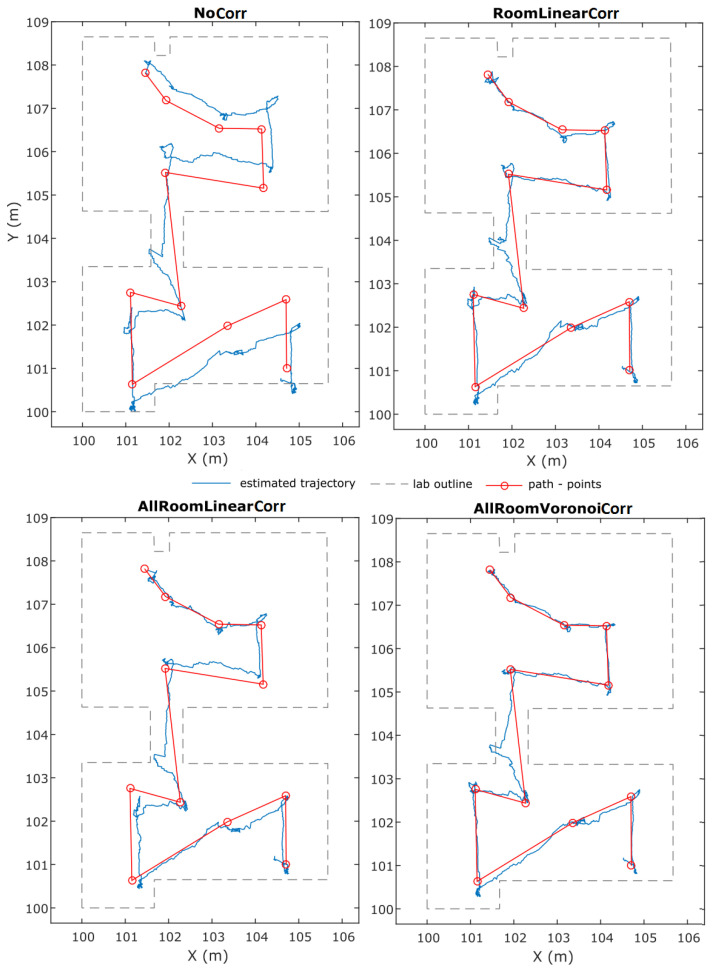
Kinematic trajectories generated using UWB ranging and the alternative correction methods in campaign 1.

**Figure 26 sensors-24-07520-f026:**
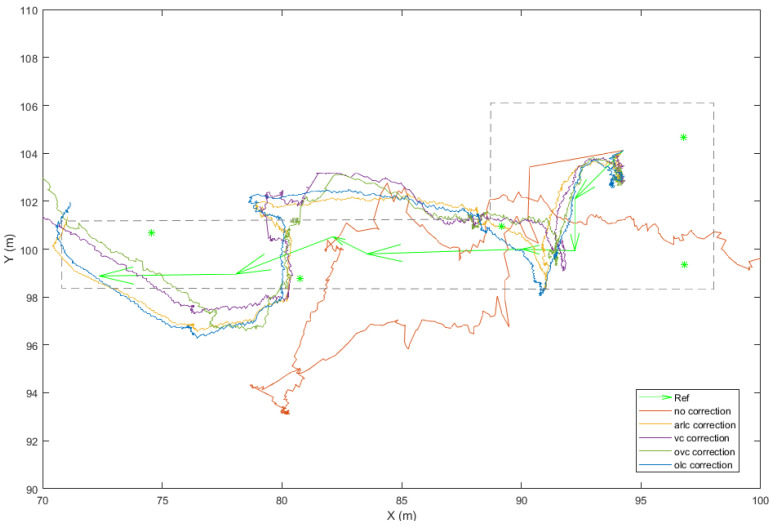
Kinematic trajectories obtained using Wi-Fi RTT ranging for the different correction methods for scenario in campaign 2. With green "*" are denoted the WiFi RTT APs, whereas the dashed line shows the experiment area perimeter.

**Figure 27 sensors-24-07520-f027:**
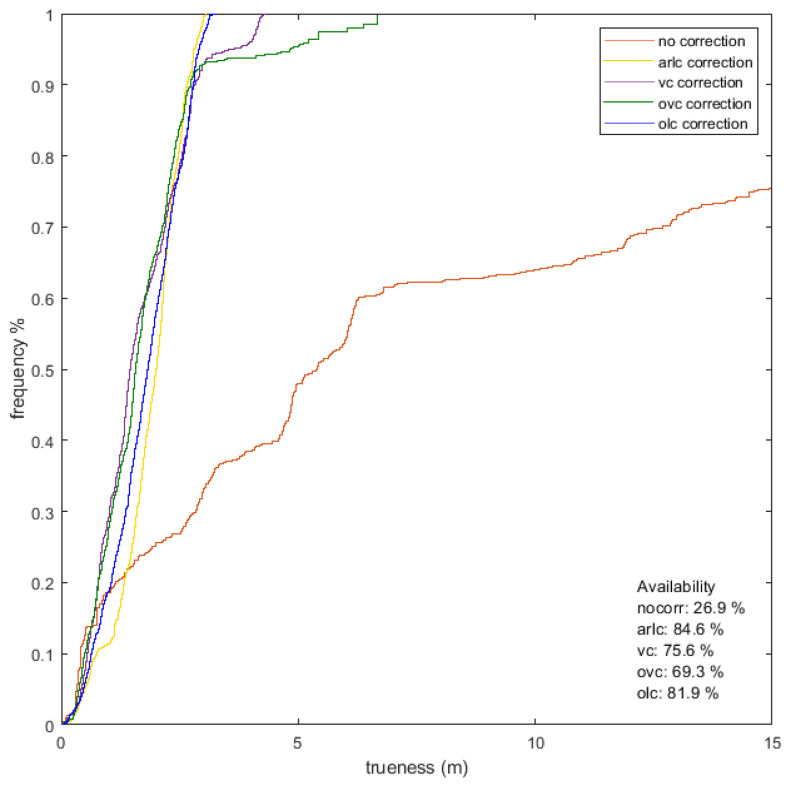
ECDF graph of position trueness using Wi-Fi RTT ranging for the different correction models for Scenario 1 in campaign 2.

**Figure 28 sensors-24-07520-f028:**
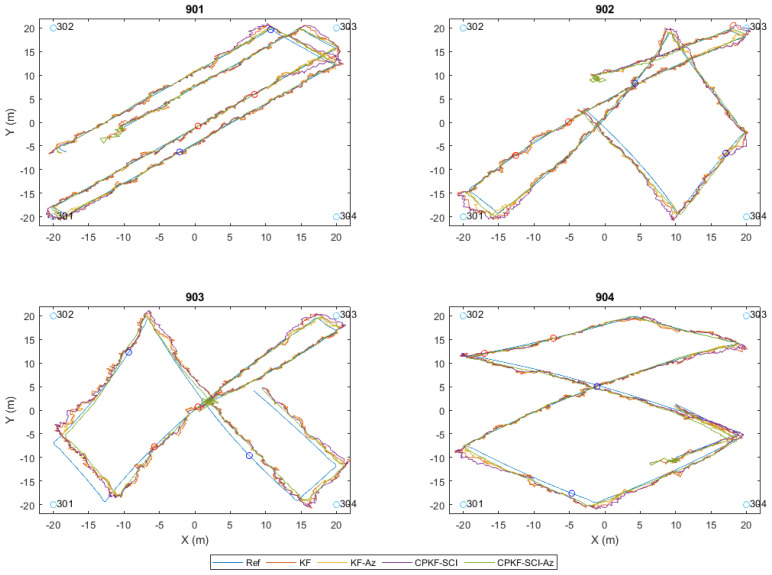
Rover trajectories for a four-rover setup applying P2I WiFi-RTT, P2P UWB ranges, and the azimuth of campaign 3 with no anchor loss, utilizing simulated data.

**Figure 29 sensors-24-07520-f029:**
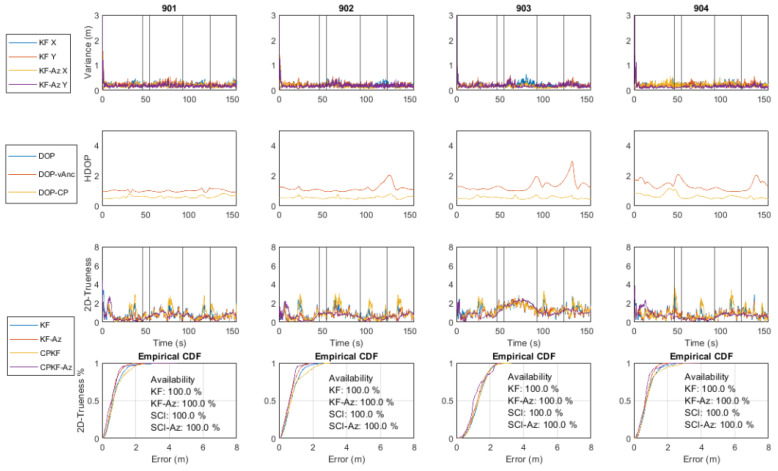
Performance quality metrics graphic summary for the generated trajectories of campaign 3 with no anchor loss, utilizing simulated data.

**Figure 30 sensors-24-07520-f030:**
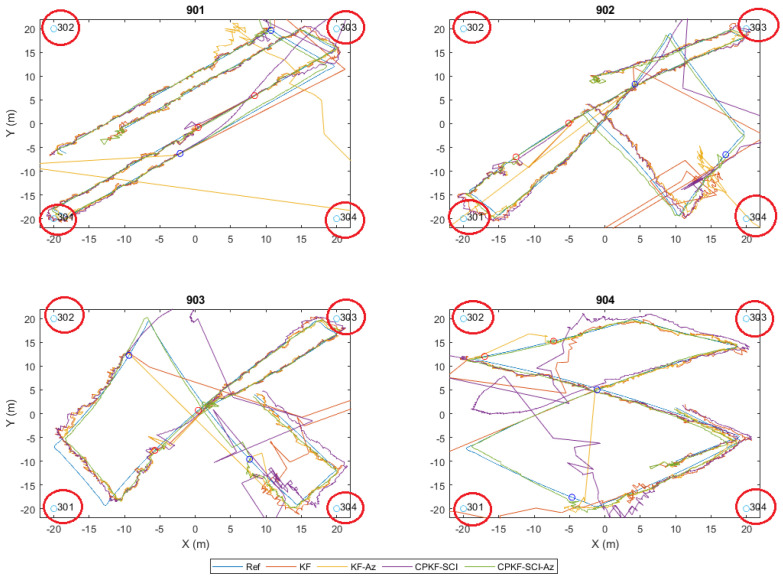
Rover trajectories obtained for a four-rover setup applying P2I WiFi-RTT and P2P UWB ranges, and the azimuth of campaign 3 with complete anchor loss. Varying anchors are highlighted with a red circle.

**Figure 31 sensors-24-07520-f031:**
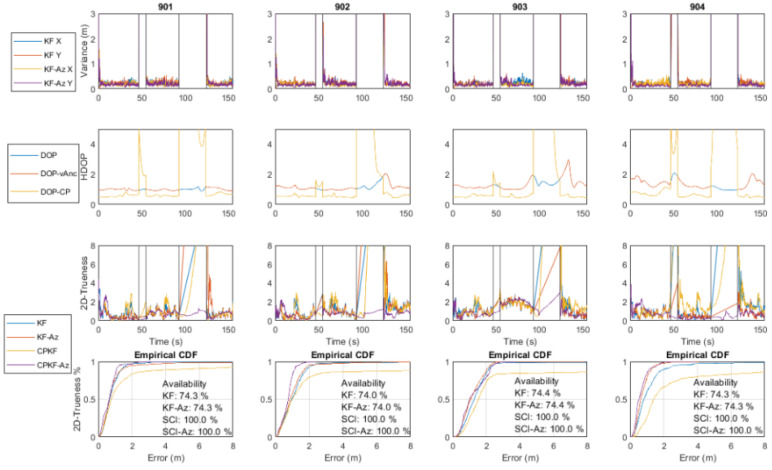
Performance quality metrics graphic summary for the generated trajectories of campaign 3 with complete anchor loss, utilizing simulated data.

**Figure 32 sensors-24-07520-f032:**
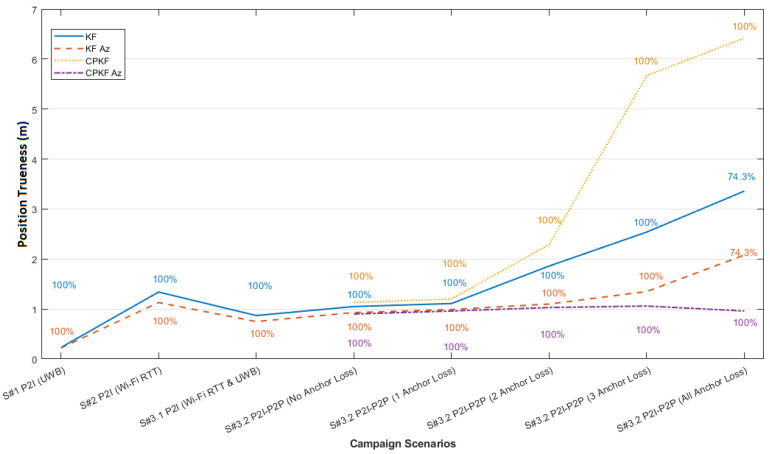
Statistical summary of positioning algorithms performance obtained for the simulation-based campaigns’ scenarios.

**Table 1 sensors-24-07520-t001:** Comparison between centralized and distributed localization architectures.

Approach	Strength	Weakness
**Centralized**	high accuracy precise node states correlation estimation	low robustness (central processor failure is critical) high communication and processing requirements not easily scalable
**Distributed**	do not require high performance central processor do not require constant network-wide communications scalable	challenging estimation of nodes correlation low accuracy in principle

**Table 2 sensors-24-07520-t002:** Comparison of distributed CP algorithms.

Algorithm	Strength	Weakness
**EKF**	Low processing requirements Fast solution computation	Assumes Gaussian distribution in uncertainty of state transition and measurement affecting accuracy
**PF**	High accuracy Can operate with non-Gaussian distributions of state transition/measurement uncertainties	High computational complexity (processing requirements) Slow computation
**SPAWN**	It is by principle a collaborative approach Good approximation of the state (under conditions)	Prone to divergence in cases of large state size Optimal for simulated scenarios but diverges in real life examples Prone to divergence when implemented on loopy-networks
**CIF/SCIF**	Incorporates cross-correlation in errors between collaborating nodes Can be implemented as EKF Suitable for real-time positioning	Mainly implemented for measurements of relative position between nodes (range-only positioning has to solve non-linearity problem)

**Table 3 sensors-24-07520-t003:** Statistical summary of range correction models obtained for the pedestrian trajectory in campaign 1 using UWB.

		Trueness [m]	
**Method**	**Mean**	**SD**	**Max**
**No Correction**	0.35	0.20	0.85
**Room Linear**	0.13	0.21	0.62
**All Room Linear**	0.13	0.08	0.49
**Voronoi Correction**	0.09	0.09	0.69

**Table 4 sensors-24-07520-t004:** Statistical summary of range correction models obtained for the pedestrian trajectory in campaign 1 using Wi-Fi RTT.

Method	Trueness Mean [m]	Trueness SD [m]	Mean Availability [%]
**No Correction**	5.08	4.32	21.8
**All Room Linear**	2.29	0.79	83.0
**Oriented Linear**	2.36	0.89	81.8
**Voronoi Correction**	10.39	6.76	55.3
**Oriented Voronoi Correction**	12.01	8.70	49.4

## Data Availability

The data are available upon request.

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
