# Peer review of "Development of Advanced Positioning Techniques of UWB/Wi-Fi RTT Ranging for Personal Mobility Applications"

_sensors, 2024, doi:10.3390/s24237520_

Round 1

Reviewer 1 Report

Comments and Suggestions for Authors

This paper develops a UWB/Wi-Fi RTT ranging and positioning technique for personal mobility in the field of UWB indoor positioning. The article is clear, reasonably divided into articles, clearly diagrammed, and based on analyzing and evaluating the existing algorithms, the design of simulation experiments, and practical experiments for verification, which demonstrates the results of a sufficient amount of scientific work. However, I have the following suggestions:

1. Please add some necessary theoretical formulas.

2. How is the performance of some of the significant intervals in the method proposed in this article and the comparison with other methods?

3. What are the limitations of this article?

4. Please add the result evaluation methods such as RMSE.

5. Check and verify the correctness of the textual content, the format of the article, the grammar, and the format of the references and the citation, such as the article appeared in the more[?]

6. As the article contains a large number of illustrations, please double-check the content of the illustrations such as fonts, labels, and the meaning of different axes.

7. The article is quite lengthy, please reduce it appropriately.

Comments on the Quality of English Language

Proofread for grammatical accuracy and clarity. Ensure that every sentence contributes to understanding the paper in its entirety. To achieve a balance between technical language and accessibility to a wider audience.

Reviewer 2 Report

Comments and Suggestions for Authors

The paper titled "Development of Advanced Positioning Techniques of UWBWi-Fi RTT Ranging for Personal Mobility Applications" is a review article that explores various positioning techniques based on radio communications. The article's strength lies in its engaging research topic and extensive reference list (43 sources). However, the article lacks critical technical details regarding the radio communication systems:

  1. What is the operating frequency band of systems like Wi-Fi (e.g., 2.4 GHz or 5 GHz)? This is crucial due to potential signal interference.

  2. How does channel congestion (multiple users sharing the same channel) impact the accuracy of position detection? The scenario with only one user per channel is far from typical.

Reviewer 3 Report

Comments and Suggestions for Authors

In this paper, Perakis et al. proposed a Ultra-wide Band (UWB) and Wireless Fidelity (Wi-Fi) based on RTT (Round-Trip Time) measurements for pedestrian user localization. For that purpose several scenarios are designed either using real observation or simulated data. In addition, the localization of user groups within a neighborhood based on collaborative navigation (CP) is investigated and analyzed. The methodology applied for CP, leverages the hybrid nature of the range measurements obtained by UWB andWi-Fi RTT systems. The results obtained demonstrate that a performance improvement in respect to position trueness for UWB and Wi-Fi RTT cases of the order of 74% and 54%, respectively, is achieved due to the integration of these techniques. The proposed localization algorithm based on a P2I/P2P (Peer-to-Infrastructure/Peer-to-Peer) configuration provides a potential improvement of position trueness up to 10% for continuous anchor availability, i.e. UWB known nodes or Wi-FI access points (APs). Its full potential is evident for short duration events of complete anchor loss (P2P-only), where an improvement of up to 53% in position trueness is achieved. Overall, the performance metrics estimated based on the extensive evaluation campaigns, demonstrate the effectiveness of the proposed methodologies. Here below are some minor comments:

(1) For Fig. 1, the axis lacks title and unit.

(2) For Fig. 12, the unit for RSS should be the same for two figures.

(3) It is better for the authors to show a detailed table for trajectories generated using UWB ranging and the alternative correction methods.

(4) Some recent works in the field of metasurfaces might be helpful for personal mobility applications, for example Nature Communications, 15, 6682, (2024).

Comments on the Quality of English Language

None.

Round 2

Reviewer 1 Report

Comments and Suggestions for Authors

I have no other suggestions.

Author Response

Dear reviewer thank you once again for your initial comments and the recent acceptance response.